# Effects of Romanian Student's Awareness and Needs Regarding Plastic Waste Management

Gratiela Dana Boca [1],* and Sinan Saraçli [2],*

[1] Department of Economics, Faculty of Sciences, Technical University of Cluj Napoca, 430122 Baia Mare, Romania
[2] Department of Biostatistics, Faculty of Medicine, Balıkesir University, Balıkesir 10145, Turkey
* Correspondence: gratiela.boca@econ.utcluj.ro (G.D.B.); ssaracli@balikesir.edu.tr (S.S.)

**Abstract:** The purpose of this study is to examine the effects of needs and awareness of university students on their environmental behaviour. With this purpose the data was collected from 537 students from the University of Cluj Napoca, Romania, from the engineering and management specializations respectively via an online questionnaire. The questionnaire was structured in four parts including 29 questions in total. The first part is meant to identify the students' characteristics (gender, field of study, participation and attendance in field-specific activities, and if he/she is an environmentalist). The second part is meant to determine the students' awareness regarding plastic and plastic pollution. Another part is meant to determine the needs of students and the manner in which they learn and gather information. The last part allows the determination of the students' behavior in their daily life (use of bio plastic bags, environmental protection). The results show that students have enough information about biodegradable plastic but they act depending on the situation, respecting or not the rules for selecting plastic waste. The female student' pay a lot of attention to selecting and choosing bioplastic products. The male students are directly involved in cleaning nature. Management students pay attention to small details as compared to engineering students who choose bioplastic even though the costs are higher. Related with their thoughts the factors effecting the opinion of either they are environmentalist or not are also examined. Being aware of the plastic waste show significant effect from the sides of awareness and behaviour. Finally, the structural model show that strongest connection is between students' awareness about the plastic problem and the need to adapt to new regulations. Using the model universities can promote the importance of bioplastic through study programs or by involving students in volunteering activities, through their active involvement in environmental protection, and selective waste recycling.

**Keywords:** plastic waste; student's awareness; behavior; needs; plastic waste management

## 1. Introduction

Sustainable development has a key component to environmental issues which is one of the European Union's horizontal policies. In order to join Europe, a stand-alone place was occupied by Chapter 22 Environmental protection, and within it the position Waste was noted [1]. Alignment with European standards is a priority and a necessity in the issue of plastic waste. Romania has committed itself to a substantial reduction of waste quantities over a period of 15 years, to make the necessary systems for collection, recycling and revaluation, but also those of future protection of the environment, and a population against pollution. Statistics showed that almost a third of plastic waste is recycled in Europe. Plastic production has grown exponentially worldwide in just a few decades, from 1.5 million tonnes in 1950 to 359 million tons in 2018 [2].

After a sharp decline in the first half of 2020 due to COVID-19, plastic production recovered in the second half of the year, and with that came plastic waste [2] but EU is already taking steps to reduce the amount of plastic waste. Androniceanu et al. [3] estimated that in 2019, globally, the production and incineration of plastics released more

than 850 million tons of greenhouse gases into the atmosphere, and by 2050 these emissions could reach 2.8 billion tons. Some of these can be avoided by better recycling.

Unfortunately, Romania is among the 14 EU member state that risks not being able to meet the 50% waste recycling target in 2020. The European Commission's analysis shows that the main causes of the problem are related to the fact that [4]:

- The selective waste collection process is not sufficiently implemented;
- Insufficient incentives have been adopted to direct waste to recycling;
- Schemes on extended producer responsibility for packaging are not efficient and do not have the capacity to cover the costs of selective collection;
- There is no infrastructure to support selective collection, which makes it difficult for citizens to get involved in this process.

Romania recycles only 13% of waste, and the remaining waste disposal at landfills is 69%, among the highest in Europe [5]. Although recycling cannot replace the need to significantly reduce the amount of disposable packaging and is by no means a justification for increasing plastic production, it has an important role to play in the transition to a plastic-free economy [6].

Bastos de Sousa [7] mentioned the essential role of the Agenda 2030 for Sustainable Development [8], which sets the Sustainable Development Goals and provides a critical overview of the role of plastic, mentioning its advantages and disadvantages. The plastic waste problem must be solved, first of all, at the source. Manufacturers and traders must gradually reduce and then abandon the production of disposable plastic packaging, and invest primarily in reusable systems.

An analysis of plastic waste management in Romania, in the context of membership in the European Union, can present in the mirror the entire activity related to this field. The directions for improvement and harmonization would be to develop strategies aimed at the recovery and reuse of waste. Correctness of behavior, ensuring the limitation of waste where resources are insufficient and expensive for the growing, diversified, but also sophisticated needs of people.

Romania, for its part, faces an annual significant increase of the amounts of waste, which causes problems related to its storage, recycling, recovery, or destruction. The huge amounts of waste result from human or industrial activities with different forms of impact: changing the landscape, visual discomfort, air pollution, changing soil fertility, and so on. Romania must involve—without exception—all institutions and every citizen in this action and spend whatever it takes to get a clean, glowing face [4].

Romania must involve—without exception—all institutions and every citizen in this action and spend whatever will be necessary to get a clean and shiny face. In the context of integration into the European Union, our country was obliged to organize the entire activity related to this field, to develop strategies aimed at the valorization and reuse of waste. Correctness of behavior in this sense is essential because it will ensure the limitation of waste in conditions where resources are insufficient and expensive for the growing, diversified, but also sophisticated needs of Romanians [5].

Currently, Romania gives detectable signals, but insufficiently intense to join not only declaratively, but also de facto the initiative of sustainable global development. Alignment with European standards is a priority and a necessity. This is the legacy caused by a policy of forced industrialization focused only on the results without taking into account the consequences that we are only seeing today [6]. It is a long way, and it takes time to repair and build a framework on the go that involves economic and social responsibility in accordance with the rules of the European Union. It is extremely difficult for a country whose population does not yet have an education regarding environmental protection and adequate recycling. In this sense, the signals that are transmitted refer to the practical change from the roots of comfortable, wasteful, and indifferent behavior towards the environment.

## 2. Literature Review on Plastic Waste Management

Researchers from different countries have taken the plastic problem seriously and have identified until now a few factors which can influence waste management: knowledge, needs, attitude, behavior and awareness, and have established similarities between students' awareness regarding the environment, similar behavior, and needs to increase students' knowledge of environmental education.

The diversity of fields (education, medicine, engineering, services) in which plastic is used was subjected to research in order to identify the behavior related to plastic waste all over the world. Education was a priority for researchers considering that today's students are tomorrow's citizens. A study conducted by Yusuf and Fajri [9] in Indonesia and Liao and Li [10] in China highlight the differences in students' behavior, engagement, and environmental knowledge on waste management in correlation with their major in sciences or social sciences through the campus programs and environmental education and knowledge regarding the separation of solid waste on campuses.

The behavior of students in environmental sciences obtained a higher score in comparison with the students from social sciences, possibly because they have more information and their curricula is more oriented towards environment topics. Situmorang et al. [11] compared Taiwan students' knowledge and behavior on plastic waste with knowledge on the negative impact of plastic waste. The results show that environmental education at undergraduate level can increase students' awareness in particularly the plastic waste problems. As a conclusion, [11] one solution for plastic waste is to increase students' awareness of environmental education and initiate participatory environmental programs.

Through research, Harman and Yenikalayci [12] also determined that students are aware of the effects of educational activities on the recovery, reuse, recycling, or use of plastic bags or dishware [13], and zero waste practices in waste management. But they concluded that individual behavior has an important role in waste management, yet most do not know the basics of waste management. Uehara et al. [14] in their study showed that by learning the rules of plastic waste separation, campus students improved their plastic separation behavior. Bennett et al. [15] sustain that educating students about plastic recycling will give them the knowledge and skills they need for their future as consumers and to implement plastic recycling systems professionally.

Nyavor-Akporyo et al. [16] mention the benefits of recycling plastic in campus by implementing course training programme not only for students but also for academic staff. In their research among Nigerian university students, Aikowe and Mazancová [17] study students' plastic waste sorting intentions. They [17] highlight the importance of other influencing factors such as environmental awareness, volunteering, and study program in evaluating plastic waste sorting intentions.

Dalu et al. [18] pay attention to plastic pollution, and results indicate that environmental programs can help students and citizens in better understanding of the interactions between human activities and the natural environment. Kaffashi and Shamsudin [19] mention that it is also important to transform planned behavior of citizens regarding pollution. Islam et al. [20] from Sydney Australia pay attention to e-waste focusing on young university students' awareness, perception, and disposal patterns for electrical waste and electronic equipment. They [20] mention that the results of the study showed that although the consumers were aware of what electronic waste (e-waste) is, there is a severe lack of knowledge regarding collection points and current recycling programs indicating that awareness programs are essential to avoid the incorrect disposal of electronic elements.

We can see that all over the world the problem of plastic waste is at the center of attention and university studies have tried to identify the behavior and attitude of students, in our case towards plastic, in order to find solutions.

So students from different countries are facing the same problems regarding plastic waste management. Some weak points were discovered by Owojori et al. [21] and their results show that students' knowledge from South Africa on waste management is low and inadequate because students are willing to partake in recycling projects to improve but

also some of the students require motivation to participate in recycling schemes through economic incentives.

Árnadóttir et al. [22] mention Holland students' positive attitude in university cafeterias and their willingness to behave pro-environmentally, but he identifies students' gap between intention and behavior. Bashir et al. [23] investigated the awareness of Malaysian students through the program introduced in the hostel area of university students to measure students' attitude and practice towards waste separation and recycling.

A significant effect of the pandemic on plastic waste generation and management was registered. Prasetiawan and Wasisto [24] investigated the relationship between students' perception and attitudes in waste management using the Internet leads to changes in attitude and behavior in society. Several studies have also highlighted the positive impact of COVID-19 on the environment, while there are limited published studies on the long-term negative impact of COVID-19 on the environment and waste management. A negative aspect was observed by Ranjbari et al. [25] and Mohamed et al. [26] because during the COVID-19 pandemic there was a significant increase in the demand for personal protective equipment, face masks, increasing globally the amount of plastic waste in the health sector. The effects of the COVID-19 pandemic were also felt on the production, use, and waste of plastic materials, in economic activities and the use of plastic materials. The use of plastic materials decreased in the industrial sectors, as did the consumption of plastic materials in construction and the automotive industry. The pandemic has led to an increased demand for single-use plastic, increasing the pressure on this already out-of-control problem. Peng et al. [27] noted that plastic waste also harms marine life and has become a major environmental concern worldwide.

*Students' Attitude and Behavior Regarding Plastic Waste*

To identify the relation between different factors which can influence students' behavior, attitude, and knowledge regarding plastic waste management statistical tools were used.

Kaushik Dowarahey et al. [28] apply a preliminary survey to assess the awareness, attitudes, behaviors, and opinions pertaining to plastic and microplastic pollution among students in India. In the Henan institute from China, Qu et al. [29] investigated the students' attitude, as well as behavior regarding waste separation. The results presented that knowledge can influence the students' behavior and also attitude can influence in a positive way the students' behavior. Ilić-Živojinović et al. [30] identify medical students behavior and attitude regarding waste.

Victorelli et al. [31] mention in their article that the theoretical learning of students in Brazil has improved, encouraging them to properly separate and package waste. Ferdous and Das [32] in their investigation try to understand and measure students' attitude for the use of plastic materials. They identified a gap between knowledge and behavior that required attitude building and behavior change with the help of the academic environment. Results obtained [32] about plastic pollution understanding and students' perceptions indicate that plastic pollution has been integrated into the school curriculum in technology, natural sciences, geography, life science, life skills and life orientation subjects. Situmorang et al. [11], relating students' behaviors and attitude showed that they are involved in purchasing products with plastic packaging, preparing shopping bag, re-using plastic bags. The study found the positive impact between environmental knowledge on plastic waste and students' behavior. Aikowe and Mazancová [17] findings from their study of plastic waste sorting in Nigeria suggest a welcome prospective on the relevance of introducing waste sorting management practices such as recycling bins in university and campaigns for waste sorting and recycling activities. Fan et al. [33] identify the motivation–intention–behavior' model on waste sorting in China and Singapore. Khan et al. [34,35] and Colorado et al. [36] investigate the behaviors of organizations towards a circular economy for materials (plastics).

Only in recent years, a special attention was given to plastic recycling and reuse into new products such as clothing, bottles, carpets, etc. in order to prevent waste generation. Wang et al. [37] study intention to use recyclable packaging in consumers' behavior.

Shevchenko et al. [38] pay attention to bio-based products as alternatives to conventional plastics to enable circular economy as alternatives to petroleum-based plastics. The solutions can be local startup programs, innovative solutions in plastic industry toward creating circularity across the supply chain. In recent years, biodegradable and bio-based plastic production and applications have been increasingly investigated by research communities and environmental science domains.

Moshood et al. [39,40] mention a new biodegradable plastic application towards sustainability, innovations in the green product. New problem or solution to solve the global plastic pollution can be sustainability of biodegradable plastics [41]. Nevertheless, the research in this area is still in its infancy stage. Qin et al. [42] pay attention to the transfer from biodegradable plastics to biodegradable microplastics, taking in consideration a new ecological factor that threatens soil environment. Peixoto et al. [43] also highlight the microplastic pollution in commercial salt for human consumption. Tamburini et al. [44] approached the plastic to bioplastic and aluminum bottles and which is the most sustainable choice for drinking water.

Eliminating plastic waste needs time and changing human mentality. Ayşe et al. [45] in their research regarding reducing plastic waste made an analysis of influences on behavior and interventions. Kumar et al. [46] emphasized that it is necessary to involve the community in the problem of plastic waste through different policies to ban plastic, but firstly, to raise plastic awareness among people. Yusuf and Fajri [9] and Opeolu [47] show that plastic waste and plastic pollution is a global concern that must be addressed collectively, and here we can emphasize the role of universities, as the nursery of future employees or leaders who will make the decisions for us in the future.

Singh Chauhan and Punia [48] mention that education may also play an important role in changing consumer behavior and attitudes toward plastic waste management. Sandu et al. [49] sustain that educating society about plastic waste will provide people with awareness about the issue and the appropriate actions to be taken, and will motivate the people to help the environment.

Education [48,49] has a key role in shaping a responsible behavior of the future adults. Environmental deterioration of biodegradable, nonbiodegradable, compostable, and conventional plastic carrier bags in the sea and soil were studied by Dilkes-Hoffman et al. [50], while Napper and Thompson [51] mention the role of biodegradable plastic in solving plastic waste. Environmental practice was under observation by Barros et al. [52] together with other factors which influence students' awareness, volunteering, and study program in plastic waste sorting intentions. These factors were found to be significant in sustainability in plastic use for the new consumption, the new "waste generation". Liao and Li [10] or Babayemi et al. [53] agree with the concept of waste generation.

Another new term of plastic waste era and environmental impact was introduced by Van Rensburg et al. [54] who studied the changes in society, but also the behavior of consumers regarding the use of plastic materials, a challenge for understanding the social perceptions of users.

Environmental sustainability and plastic waste have become a growing concern for academics and students, as universities can play a key role in building a sustainable society. Dagiliute and Liobikiene [55] and Fissi et al. [56] pay attention to another factor which can contribute to environmental sustainability challenges and opportunities for a new concept, i.e., that of "green university" (in our case the example of the University of Florence). A green university implements sustainability and awareness-raising activities, it needs to involve the scientific community and also society toward the reduction of plastic pollution and highlighting how education can improve its management by awareness in society through different teaching methods [57].



Taking in consideration that literature has so far focused on specific aspects of plastic waste and sustainability in the higher education sector, the aim of this study is to explore the impact on the Romanian students.

Taking into account the research carried out in the field of plastic waste, the authors try to identify and create a model for Romanian students but depending on their awareness, needs, and behaviors. The data will be used to identify the ways in which the university can come with information packages to attract the young generation to protect the environment from plastic waste.

## 3. Research Methodology

### 3.1. Materials and Methods

The purpose of the study was to understand the students' behavior and awareness regarding the sustainable environment and the students' behavior and knowledge with reference to the phenomenon of plastic recycling. A total of 537 students were involved in an experimental study to identify information and the level of knowledge with reference to the current problem of plastic and the need to replace it with bio plastic.

The questionnaire was applied online between June and July 2022 at the Technical University of Cluj Napoca, Romania, within the department of economics and engineering. The survey was structured in three parts:

Part 1. Students' characteristics (gender, faculty, field of study, participation and attendance at activities about environment protection and their environmental attitude (I1–I7 questions);

Part 2: Students' awareness (A1–A8 questions) to examine whether the young generation plays a responsible role towards plastic products, and their perception towards this topic and the new biodegradable plastic;

Part 3. Students' needs (N1–N5 questions) perception of the concept of plastic sustainability and their participation in different activities regarding plastic recycling.

Part 4. Students' behavior (B1–B9 questions) in relation to healthy plastic education, how many of them select plastic, if they use ecological products.

In this study we used Google Drive form to create the survey, and apply online, data analysis and statistical processing were performed using the SPSS software package, and for students' model Rigle et al. [58], Starstedt et al. [59] and Hair et al. [60] SmartPLS 3 program.

### 3.2. Sample and Measurment Tool

To measure the student's behavior, needs and awareness of plastic waste, a Likert-scale type questionnaire, ranging from 1 'Totally Appropriate' to 5 'Not at all appropriate', was applied. The research model is given in Figure 1; we took into consideration: N—Needs; A—awareness about environment and plastic, and B—Behavior.

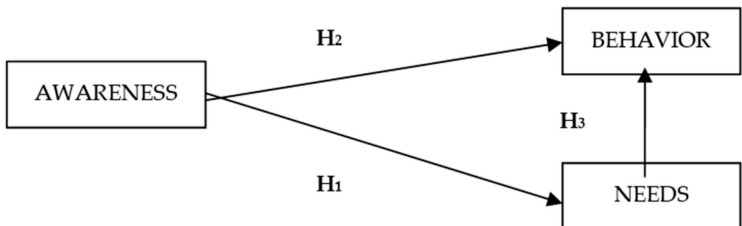

**Figure 1.** Research model regarding plastic waste. Source: By authors.

In this study, the following factors were taken in consideration:

B—students behavior (B1–B9 questions);

A—students awareness regarding plastic waste (A1–A8 questions);

N—students needs (N1–N5 questions).

To determine the dimensions of students' behavior and identify students' needs and awareness about plastic waste and bioplastic, an Explanatory Factor Analysis (EFA) was applied to the data set. By independent samples *t*-test, the hypotheses were tested using the SPSS statistical analyses software IBM SPSS Statistic 25.

The hypotheses for the model given in Figure 1 are as below:

**H₁:** *Students' awareness towards plastic waste effect their needs.*

**H₂:** *Students' awareness towards plastic waste effect their behavior.*

**H₃:** *Students' needs toward plastic waste effect their behavior.*

*3.3. Evaluation of the Measurement Model*

In SEM, latent variables are measured with two different methods as formative and reflective. The factor analysis, validity and reliability results for the measurement model in the study are given in Table 1. Formative models show the relationship from the items to the latent structure, while the reflective models show the relationship from the latent structure to the items.

**Table 1.** Factor Analysis, Validity and Reliability Results.

| Factor/Item | Factor Loading | Cronbach' α | rho_A | CR | AVE |
|---|---|---|---|---|---|
| *Awareness* | | 0.941 | 0.943 | 0.951 | 0.710 |
| *A1* | *0.830* | | | | |
| *A2* | *0.826* | | | | |
| *A3* | *0.715* | | | | |
| *A4* | *0.888* | | | | |
| *A5* | *0.829* | | | | |
| *A6* | *0.873* | | | | |
| *A7* | *0.891* | | | | |
| *A8* | *0.874* | | | | |
| *Needs* | | 0.922 | 0.923 | 0.942 | 0.766 |
| *N1* | 0.759 | | | | |
| *N2* | 0.894 | | | | |
| *N3* | 0.905 | | | | |
| *N4* | 0.912 | | | | |
| *N5* | 0.897 | | | | |
| *Behavior* | | 0.924 | 0.933 | 0.937 | 0.627 |
| *B1* | 0.691 | | | | |
| *B2* | 0.740 | | | | |
| *B3* | 0.853 | | | | |
| *B4* | 0.833 | | | | |
| *B5* | 0.820 | | | | |
| *B6* | 0.858 | | | | |
| *B7* | 0.854 | | | | |
| *B8* | 0.638 | | | | |
| *B9* | 0.808 | | | | |

The analysis results in which the cross-loading were evaluated are shown in Table 2. According to Table 2, it was determined that the loads of each structure's own items were higher than the cross-loads of other structures.

**Table 2.** Results of Cross-Loading Analysis.

| Factor/Item | Awareness | Needs | Behavior |
|---|---|---|---|
| *Awareness* | | | |
| A1 | 0.830 | 0.748 | 0.830 |
| A2 | 0.826 | 0.716 | 0.812 |
| A3 | 0.715 | 0.643 | 0.702 |
| A4 | 0.888 | 0.838 | 0.737 |
| A5 | 0.829 | 0.717 | 0.718 |
| A6 | 0.873 | 0.829 | 0.701 |
| A7 | 0.891 | 0.851 | 0.732 |
| A8 | 0.874 | 0.826 | 0.701 |
| *Needs* | | | |
| N1 | 0.731 | 0.759 | 0.748 |
| N2 | 0.814 | 0.894 | 0.703 |
| N3 | 0.823 | 0.905 | 0.723 |
| N4 | 0.828 | 0.912 | 0.715 |
| N5 | 0.813 | 0.897 | 0.697 |
| *Behavior* | | | |
| B1 | 0.582 | 0.544 | 0.691 |
| B2 | 0.610 | 0.548 | 0.740 |
| B3 | 0.767 | 0.710 | 0.853 |
| B4 | 0.724 | 0.669 | 0.833 |
| B5 | 0.729 | 0.695 | 0.820 |
| B6 | 0.772 | 0.725 | 0.858 |
| B7 | 0.783 | 0.759 | 0.854 |
| B8 | 0.530 | 0.462 | 0.638 |
| B9 | 0.726 | 0.666 | 0.808 |

These results indicate that discriminant validity is acceptable. In the evaluation of reflective measurement models, Composite Reliability (CR), indicator reliability, convergent validity, Average Variance Extracted (AVE), and discriminant validity are examined and cross-loadings are evaluated to ensure discriminant validity [54].

In terms of internal consistency, composite reliability and Cronbach's $\alpha$ value should be higher than 0.70. An AVE value of 0.50 and above indicates sufficient convergent validity.

The fact that each of the factor loads is greater than 0.708 indicates that the reliability of the indicator is provided. If the loads are below 0.70, it is recommended to examine the composite reliability value when the item is removed, instead of removing the items [59]. If factor loading are lower than 0.40, it is recommended to remove items from reflective scales.

According to Table 1, Cronbach's alpha, rho_A and composite reliability values were found to be over 0.70, and the AVE values of the scales were found to be high at 0.50. After reliability analysis, exploratory factor analysis (EFA) was applied, and the structural equation model was developed after removing items which had factor loading lower than 0.5.

## 4. Results

### 4.1. Students Characteristics

The analysis carried out from the point of view of the students, whether they are ecologists in 36.69% or not in 63.31%, gave the following results, taking into account gender, as in Table 3. It is very important also to mention that only 24.58% of the students attend a conference on nature conservation or about plastic issues (Table 3). Conversely a majority of 75.42% of students did not attend any conference or any activity about the protection of the environment and nature regarding plastic waste. Boys are more involved, i.e.to a percentage of 13.22%. Also 71.88% of them mention that they never take part in environmental activities organized within the university, and only 28.1% participate in such activities organized by the university, of which again boys are more active, to a percentage of 18.06%. 36.69% of students consider themselves environmentalists, of which 17.32% are girls and a majority of 53.45% consider themselves environmentalists only sometimes depending on the situation.

**Table 3.** Students' individual characteristics regarding active participation.

| Gender | | | F | M | Total |
|---|---|---|---|---|---|
| Have you attended a conference on nature conservation before? | Yes | | 61 | 71 | 132 |
| | No | | 172 | 233 | 405 |
| | | Total | 233 | 304 | 537 |
| Do you think you are an environmentalist? | Yes | | 93 | 104 | 197 |
| | No | | 25 | 28 | 53 |
| | Sometimes | | 115 | 172 | 287 |
| | | Total | 233 | 304 | 537 |
| Did you take part in environmental activities organized within the university? | Yes | | 54 | 97 | 151 |
| | No | | 178 | 207 | 386 |
| | | Total | 233 | 304 | 537 |

In conclusion, young students are comfortable and do not get involved in extracurricular activities because they spend their free time doing other activities.

Table 4 presents the correlation between students' specialization and their individual characteristics. Of the 537 respondents, 53.63% are engineering students in IT, automations and electrical engineering, and 46.37% are from management and business administration. The management students admit to 38.15% that they are ecologists, and 51.81% admit to being ecologists only sometimes, i.e., when they remember or are imposed rules that they must follow.

85.85% of the students from both fields of study have information related to biodegradable plastic and the benefits of its use. A majority of 87.95% was obtained by the management students. Although theoretically they are better informed than those from engineering, in practice they do not participate in the environmental activities organized by the university in a massive percentage of 91.97%. When asked about participation in conferences about nature conservation, a percentage of 79.91% of them stated that they do not participate, citing that their profile refers to the economic side of bio plastic.

We can conclude that students are not involved in collateral activities with their field of study, but as individuals some of them apply the rules of where plastic waste comes from. Students in both courses agree with the rules and special selection of plastic waste and prefer to use low-degradation items. Boys are more involved in practical activities related to nature, or participate in actions organized by the university. The girls have the information about bio plastic but prefer not to be actively involved.

**Table 4.** Students' participation depending on their field of study.

| | | Automation | IT | Management/Business | Electrical | Total |
|---|---|---|---|---|---|---|
| | | | | **In Which Department Do You Study?** | | **Total** |
| Did you take part in environmental activities organized within the university? | Yes | 48 | 42 | 20 | 41 | 151 |
| | No | 66 | 47 | 229 | 45 | 386 |
| | Total | 114 | 89 | 249 | 85 | 537 |
| Have you attended a conference on nature conservation before? | Yes | 26 | 25 | 50 | 31 | 132 |
| | No | 88 | 64 | 199 | 54 | 405 |
| | Total | 114 | 89 | 249 | 85 | 537 |
| I know that petroleum product (plastics) take a long time to biodegrade? | Yes | 96 | 79 | 219 | 67 | 461 |
| | No | 7 | 4 | 10 | 6 | 27 |
| | No idea | 11 | 6 | 20 | 12 | 49 |
| | Total | 114 | 89 | 249 | 85 | 537 |
| Do you think you are an environmentalist? | Yes | 48 | 24 | 95 | 30 | 197 |
| | No | 9 | 13 | 25 | 6 | 53 |
| | Sometimes | 57 | 52 | 129 | 49 | 287 |
| | Total | 114 | 89 | 249 | 85 | 537 |

So the question that arises is how to attract the younger generation to be more active and involved in this issue.

In order to create the cross model and find solutions, we also studied the correlation between the students' awareness and behavior depending on their field of study and gender to obtain more information.

*4.2. Students Needs*

Teddy Prasetiawan and Wasisto [24] sustain that education about waste in schools plays a significant role in increasing students' perceptions of the environment. Previous research has suggested that students' perceptions of plastic waste management can positively impact the future generations' preparation as agents that drive changes in attitude and behavior in society. Table 5 presents the results obtain for students' needs items N1–N5.

**Table 5.** Items for students' needs regarding bioplastic.

| | |
|---|---|
| N1 | I would like to get new information about the use of bioplastic products. |
| N2 | Bioplastics should replace conventional polymers in the future. |
| N3 | I think that all kinds of studies on the pollution of traditional plastic products should be increased. |
| N4 | I think that bioplastics should be used in mass social events. |
| N5 | I think that bioplastics should be used in takeaway products. |

Maybe 43.4% of the students are aware that this process requires a long time and tailor-made strategies.

56.6% of the students do not want to get new information about bioplastic, maybe due to the fact that their field of study is different they consider that they know enough or the information they have through mass media promotion is enough. In a percent of 13.2% some students are still aware of the importance of plastic waste and consider it is necessary to continue research in this field of removing plastic and replacing it in time.

For item N1: "*I would like to get new information about the use of bioplastic products.*" only 20.3% percent of the students from both fields of study are open to learn and to obtain new information about bioplastic, maybe because some of them are already working, or they do not want to spend extra time with new information. Here they mention qualified people who are already payed to do a specific job.

For item N2: "*Bioplastics should replace conventional polymers in the future*" 30.5% of the students agree and totally agree that in the future plastic must be replaced with the new biodegradable material, to prevent pollution.

But for item N4: "*I think that bioplastics should be used in mass social events*" the percent increases to 31.1%, so students know about the plastic waste especially from social events in which they are involved.

For item N5: "*I think that bioplastics should be used in takeaway products.*" The highest value of 31.5% of the students agree with the bioplastic takeaway maybe because they know the importance to protect the environment and they have an environmental education already.

In conclusion students have information, knowledge about plastic and plastic waste, they already apply and are oriented in their daily life towards the use of the new bioplastic material, but they do not want to be involved in courses or lectures on that specific topic in their extra curricula time.

### 4.3. Students' Awareness

From the point of view of awareness, the results for items A1–A8 from Table 6 show that students are informed and they are concerned about the issue of plastic waste and plastic pollution, but they are acting individually.

**Table 6.** Awareness Items selection.

| | |
|---|---|
| A1 | I prefer products that are less harmful to nature |
| A2 | I use low-damage objects such as paper and glass. |
| A3 | I recognize the bioplastic logos on the products. |
| A4 | I prefer the products obtained from the bioplastics industry because they are renewable. |
| A5 | I prefer to avoid products that increase global warming. |
| A6 | I prefer bioplastic products because they do not harm nature when they decompose. |
| A7 | I prefer bioplastic products as they do not harm human health when degraded. |
| A8 | I prefer bioplastic products because they degrade earlier in nature. |

For item A4: "*I prefer the products obtained from the bioplastics industry because they are renewable*" the highest value of 33.9% of the students from both management and engineering prefer bioplastic products because they are renewable, that means they are informed about the plastic waste problem.

For item A5: "*I prefer to avoid products that increase global warming.*" 32% of the students agree that the plastic waste problem can increase global warning.

For item A6 "*I prefer bioplastic products because they do not harm the nature when they decompose*" the highest value of 33% of the respondents agree with bioplastic because the material does not harm nature when they decompose.

For item A7: "*I prefer bioplastic products as they do not harm human health when degraded*" a 30.1% of the students agree and totally agree to choose the bioplastic products which do not harm health after they are degraded.

For item A8: "*I prefer bioplastic products because they degrade earlier in nature*" a value of 32.1% of the respondents have information about the bioplastic degradation, in time, and they are aware of environment protection, a percent of 26.1% does not have any opinion; they are passive.

For all items a compact value between 20–30% of the respondents just sit and watch, and consider that there are specialists paid to take care of the plastic problem.

### 4.4. Students' Behavior

Students' behavior from the point of view of their gender: behaviors were shown in purchasing products with plastic packaging, preparing shopping bags, re-using plastic bags, taking own meal box, and having food on the sites to reduce single used plastic package. Students as a target are important as these individuals will be the ones making political decisions, with regards to recycling programs in their communities.

By understanding their behavior in colleges and universities it is possible for those who design recycling program to develop and modify programs in connection to their

needs and provide information programs for their education. In Table 7 we present the students' behavior, survey items B1–B9.

**Table 7.** Items for students' behavior regarding plastic waste.

| | |
|---|---|
| B1 | I buy bioplastic products, even if they are expensive. |
| B2 | When I see plastic being burned, I report it, thinking it will cause air pollution. |
| B3 | I use bioplastic bags for my grocery shopping. |
| B4 | I make an effort to use a small number of bags in my daily life. |
| B5 | I do not throw plastic products into nature. |
| B6 | I throw the recycling products into the relevant boxes. |
| B7 | I use mesh/cloth/paper bags instead of using disposable bags while shopping. |
| B8 | I inform the people around me about the use of bioplastic products. |
| B9 | When I see plastic in the green area, I take it from there. |

For item B2: *"When I see plastic being burned, I report it, thinking it will cause air pollution".* A percent of 31.3% of the students have a positive behavior when they are involved in different plastic waste situations which can harm nature and pollute it.

For item B7: *"I use mesh/cloth/paper bags instead of using disposable bags while shopping."* 30.4% are trying to adapt their behavior to the new standards; the low value obtained might be due to them not being accustomed yet with new situation or maybe the costs are increasing a little bit.

For item B8: *"I inform the people around me about the use of bioplastic products."* 33% of students from both fields of study agree with the importance of dissemination and information regarding the use of bioplastic products.

For item B9: *"When I see plastic in the green area, I take it from there"* the highest value of 33% of the students have a protective behavior towards nature and green nature by selecting waste in specific places.

In conclusion the students' behavior as individual behavior shows us that only a low percentage of students have information about plastic waste and they have proper behavior. Results present that 67.6% of the female students have positive behavior and protective attitude and protect nature when they are using plastic.

In exchange, male students pay more attention 76.2% to the distribution of information about the use of bioplastic products. They are more careful and make a greater effort when it comes to using plastic materials for 66.7%.

At the opposite end, only 33.3% of female students can hardly refrain from using as few bags as possible in their daily life, and 13% considers it impossible to adapt to these requirements.

## 5. Discussion

### 5.1. Correlation between Students' Awareness Depending on Their Field of Study

35.5% of the engineer students are aware about plastic waste and consider that they are environmentalists, (Figure 2) of which 15.8% protect the environment and take care of nature.

A percent of 30.7% accept bioplastic products because they do not harm human health when degraded. So engineering students realized the importance of bioplastic and accept the values in comparison to management students who evaluate each aspect of the plastic waste problem.

The management students agree in 57.4% with the rules and special selection of plastic waste and they prefer to use low damage objects (Figure 3).

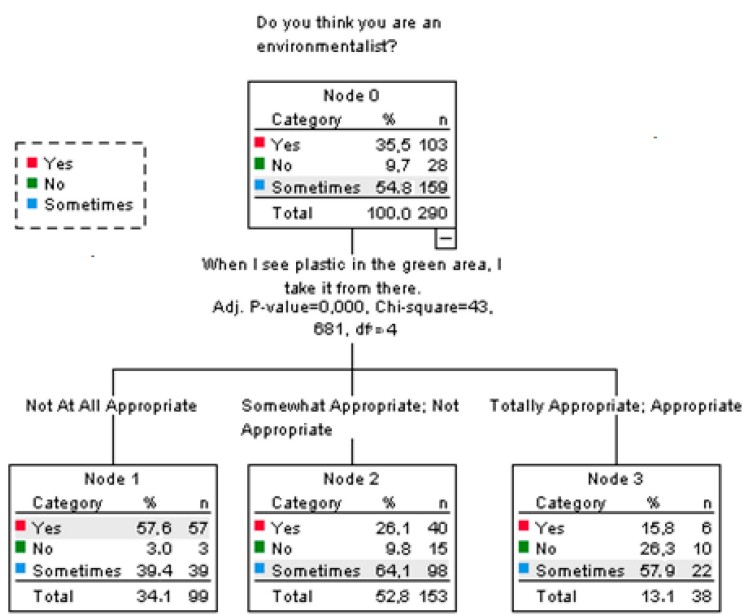

**Figure 2.** Environmental awareness for Engineering students.

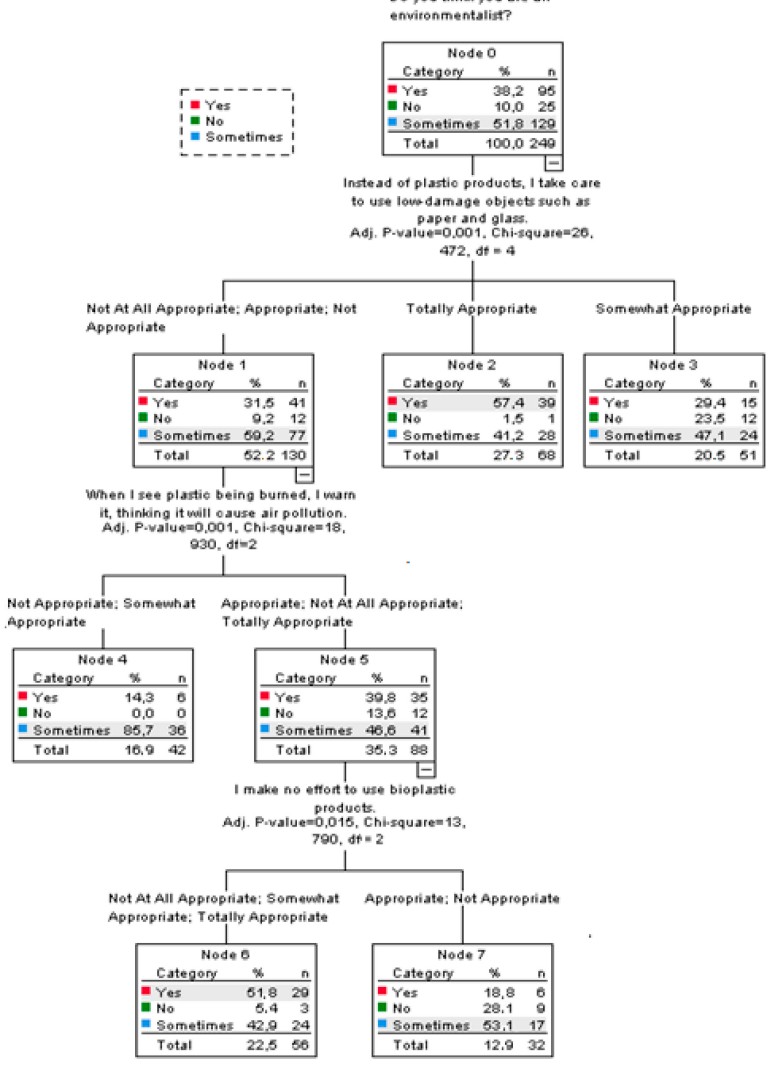

**Figure 3.** Environmental awareness for Management/Business students.

Even those who do not agree with the use of paper or glass objects in particular in a percentage of 31.5%, are aware of the importance of plastic waste. Of these, 39.9% are involved in various actions to prevent pollution, but they admit that they do not make efforts because they act individually in 51.8% in the use of the new bioplastic material. They are aware of the influence of plastic and 15.8% of engineering students have protective behavior towards nature while 57.7% do not get involved.

### 5.2. Correlation between Students' Awareness Depending on Their Gender

If we draw a parallel with the male awareness students from Figure 4 we can see that 34.2% consider themselves environmentalists. 29.7% of the male students agree to take action and keep the green nature, and clean the area from plastic objects.

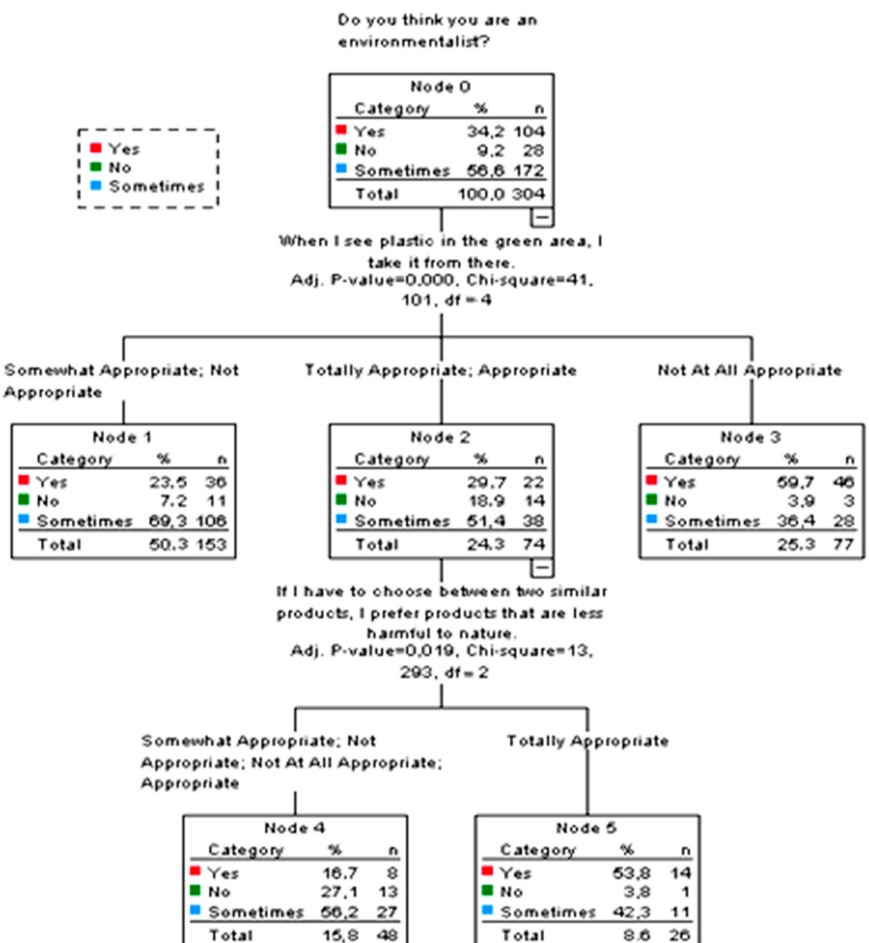

**Figure 4.** Environmental awareness of male students.

Male students get more involved when they have to take a stand and clean nature of plastic waste because they are not so sensitive and they are active in comparison with female students. 21.8% agree with the new bioplastic material even if the price is higher; of these 87.2% are male and only 12.8% are female.

To see if gender influences the students' awareness, in Figure 5 we present the results obtained for female students.

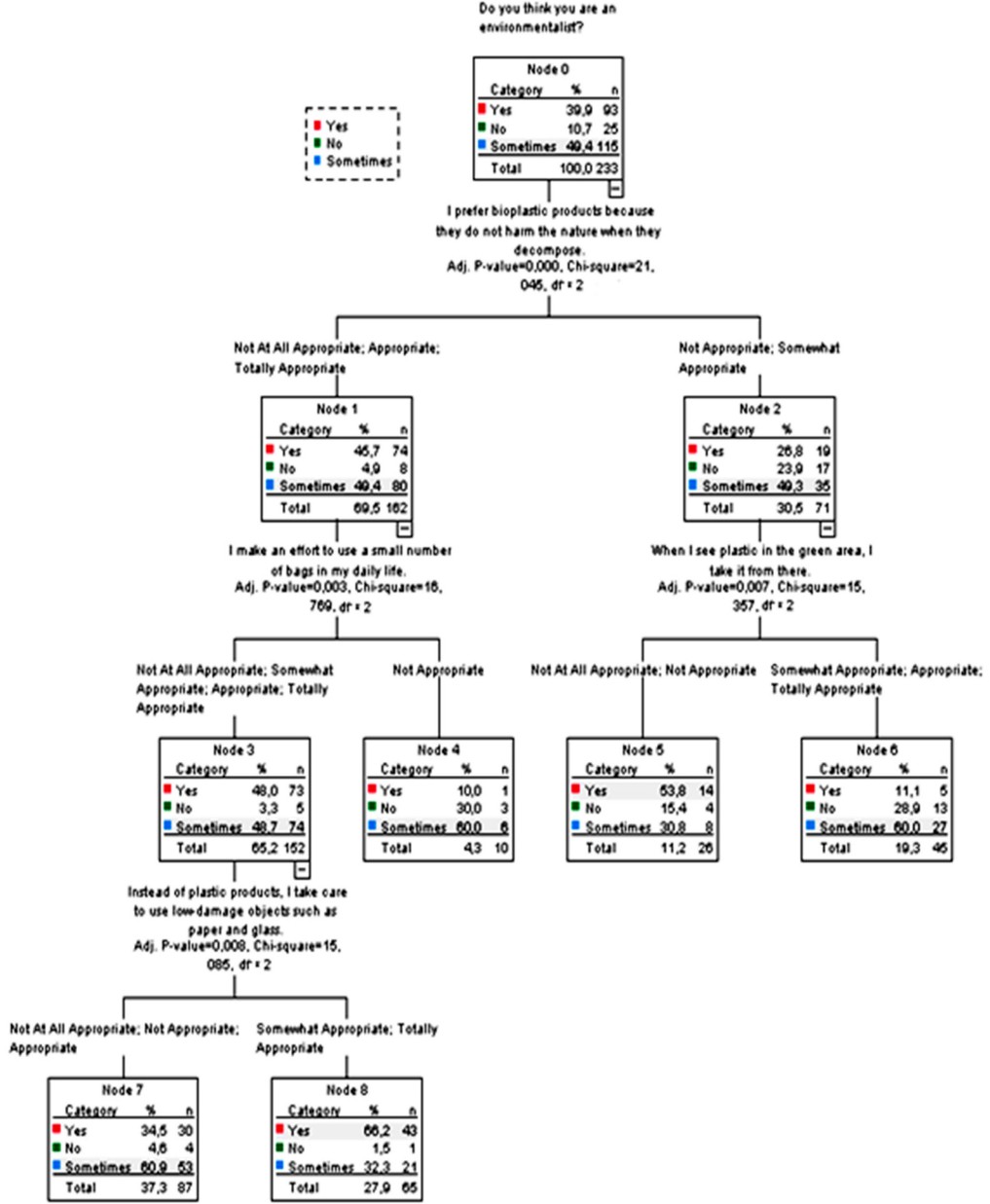

**Figure 5.** Environmental awareness of Female students.

39.9% of the female students considered themselves environmentalists, but only a low percentage of 11.1% of the female students take care of green nature and collect garbage when they see it. Female students in 48% consider that they do not make efforts regarding their daily behavior on using bags, and 66.2% of them are very careful when they use plastic objects.

In exchange 15.5% of the management students totally agree with the bioplastic because they degrade earlier in nature, the majority of which are female—82.1%. 53.8% of the male students agree to use products which are less harmful to nature a low value as compared with female students who scored 66.8%. So females are more practical when talking about using plastic products and objects.

*5.3. A Cross Model for Students' Regarding Plastic Waste*

In the evaluation of a structural model in PLS-SEM, linearity, path coefficients and significance values, $R^2$, $f^2$ and $Q^2$ values are used (Hair et al., 2017). The $R^2$ value, which shows how much of the change in the endogenous latent variables explains the exogenous

latent variables, is considered significant if 0.70, moderate if 0.50, and weak if 0.25. Higher $R^2$ value indicates the prediction accuracy of the model [57,59].

On the other hand, effect size; $f^2$ is examined in evaluating the contribution of an exogenous variable to the value of $R^2$ of an endogenous latent variable. The $f^2$ value of 0.02, 0.15 and 0.35 correspond to a weak, medium and strong effect, respectively. $Q^2$ value, which should be greater than zero, indicates the power of exogenous latent variables to predict the endogenous latent variable [60]. Results of PLS SEM for the research model are given in Figure 6.

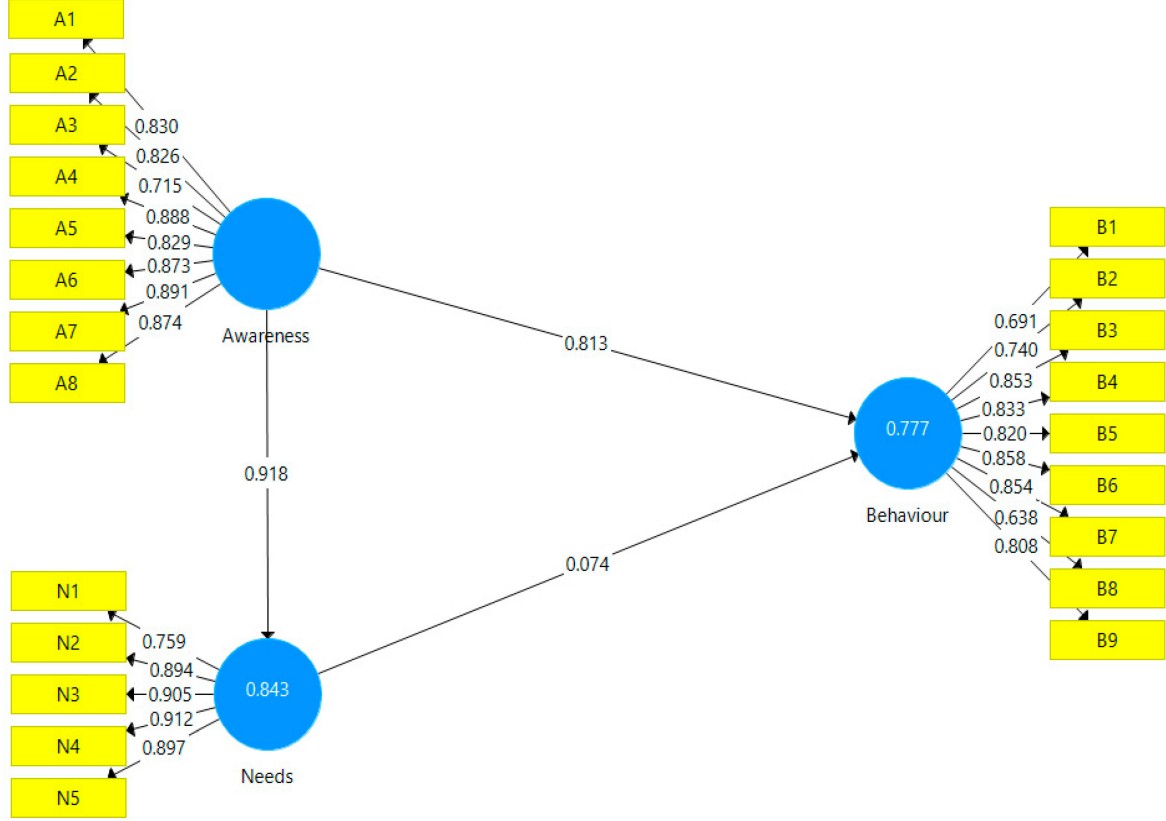

**Figure 6.** Results of the Cross model for Romanian students regarding plastic waste.

According to Figure 6 standardized path coefficients between Awareness and Needs, Awareness and Behavior and Needs and Behavior are calculated as 0.918, 0.813 and 0.074 respectively.

The significance of the path coefficients, *t*-values and *p*-values of the model are given in Table 8.

**Table 8.** Descriptive results for the research model.

| Relationship | β | *t* Value | *p* Value |
|:---:|:---:|:---:|:---:|
| A → N | 0.918 | 111.09 | 0.000 |
| A → B | 0.813 | 12.21 | 0.000 |
| N → B | 0.074 | 0.97 | 0.33 |

According to Table 7, when the *t* and *p* values of the path coefficients between the variables are examined, it can be said that while the path coefficients for A → N and A → B are statistically significant, the coefficient for N → B is not significant. These results also indicate that while the hypothesis $H_1$ and $H_2$ are accepted, $H_3$ is rejected.

As the model found statistically significant and awareness has a significant effect with the coefficient of 0.918, the item A7 "*I prefer bioplastic products as they do not harm human health when degraded*" has the highest effect on awareness with the coefficient of 0.891. Likewise N4 "*I think that bioplastics should be used in mass social events*" has the highest effect on Needs with the coefficient of 0.912 and B6 "*I throw the recycling products into the relevant boxes*" has the highest effect on awareness with the coefficient of 0.858.

In the research model in Figure 1, it was determined that the $R^2$ values of the Behaviour and Needs are 0.776, and 0.842 respectively which are well enough for the model acceptance. The effect sizes $f^2$ of Awareness on Needs, Awareness on Behavior and Needs on Behavior are also calculated as 5.350, 0.466 and 0.004 respectively. Besides these results the $Q^2$ values of Behavior and Needs was calculated as 0.48 and 0.64 respectively.

*5.4. SWOT Analysis for Romanian Students*

Plastic pollution and plastic waste is a growing global problem. But, in this endeavor, solutions must be presented for the companies involved, but also for the final consumers, for which disposable products have become a habit in recent years.

In addition, it should be emphasized that the new raw materials from which the new products will be made are wood or corn, which will lead to reduce waste. In recent decades, bottles, bags, and a wide range of plastic items have become the most toxic waste that pollute the environment.

In Romania, they are found everywhere, on top of the mountain, on watercourses, in recreational areas, or on the squares in the suburbs, floods bring pet islands to rivers and the Danube.

As we can see, this is a problem that other states also face. Based on the results obtained, university can find solutions for the students and bring added value to their first steps in implementing awareness for plastic waste.

For these reasons, there must be well-organized public consultations in which to discuss all the unclear issues indicated by the population, non-governmental organizations and the business environment.

In conclusion, for our study in Table 9 we create a Strengths, Weaknesses, Opportunities, and Threats matrix know as S.W.O.T. analysis related to understand our university strong and weak points and what we have to do for students.

**Table 9.** SWOT analysis regarding students' awareness of plastic waste.

| STRENGTHS | | WEAKNESSES |
|---|---|---|
| Universities; Student Information; Specialists in field; Interest; Mass media | B I O | No interest; Social law missing; Not observing the rules; Costs are high; Difficulties in implementation; Difficulties to follow the EU rules |
| OPPORTUNITIES | | THREATS |
| Green University Special Platform with information; Courses online; Playing by doing in the virtual world; Exchange of staff and students and of good practices Books and Virtual library; Curricula; Workshops and conferences Bilateral agreements Transfer of research ideas Dissemination of 3R's Volunteer activities Teams involve in research activities | P L A S T I C | People and their indifference; Resistance to change encouraged by keeping people; The refusal of impeachment. |

## 6. Conclusions

The results obtained after filling out the questionnaire led to the creation of the new model and identifying common elements of students' needs, students' behavior and students' awareness regarding plastic waste. For our study, Romanian students' awareness obtained the highest value so they are involved in waste plastic management.

We also verified the influence of the major where students are enrolled, in our case management students' knowledges is the highest from a theoretical point of view, but engineering students are more practical, and they are involved in activities, and participate actively in activities organized by the university, or they attend conferences on the issue of plastic.

Authors identify the similarities and differences. When we take in consideration students' gender, male are more practical in cleaning nature from plastic, in comparison with female students who pay attention to small details of using and selecting the plastic waste.

Students act individually as persons and they do not like to be involved in extra activities. They are informed from the mass media, from university, but sometimes they are lazy to follow the rules or they consider that there are qualified persons who are paid to do their job.

The results are a signal for universities which must involve students in activities with specific issues about the environmental protection and plastic waste, in volunteering activities and also in research activities to understand the importance of the issue. In Romania, efforts to combat the plastic pollution crisis are rather minor, fragmented, and generally ignore the extremely harmful consequences that both we and the environment have to bear every day.

Also on the study of bioplastic and the future vision without plastics present also that the respondents and also the organization must be careful when it comes to alternative plastics (cane or corn starch product, biodegradable and/or compostable) and analyze all the sustainability issues associated with this type of plastic.

For the compact segment identified in the research study of 20–30% of the respondents who have no opinion and do not want to be involved in any kind of activities, we can specify the following factors:

- the lack of concern for this problem, first of all by the shops and then by authorities who do not properly inform about the problem of plastic;
- lack of information is another reason specified by the respondents.

The Cross Model for students in relation to bioplastic obtained a low value, which means a weak connection between the needs of students that influence their behavior. present a weak point in the education of students regarding environmental protection, with plastic issues as a direct target. So a necessity for students, in their behavior towards sustainable environment education is influenced by knowledge.

The factors that led to this disastrous situation are the following:

- lack of a separate collection system at the source (at people's homes) on at least 4–5 types of waste: paper/cardboard, plastic/metal, glass, bio waste (food and vegetable waste) and mixed;
- lack of proper education among the population and decision-makers at central and local level;
- lack of involvement of mayors and failure on their side to assume responsibilities;
- the lack of sanctions and total lack of waste collection in rural areas, where there are no sanitation contracts, which leads most often to pollution either when disposing of them in the wild or when burning uncontrolled.

The strongest link is the one between the students' awareness; it influences their information needs regarding the plastic problem. This is where the role of universities comes into play by:

- creating platforms for courses specific to the field, creating virtual libraries as a source of useful information;

- the transfer of information between different universities or student groups for common research topics.

The students' awareness influences their behavior by getting involved in voluntary actions, through the individual actions of selecting plastic at home or at university, protecting nature by choosing biodegradable products when shopping.

*Limitation of the Study*

The case study and the creation of a model for the Romanian students constituted a challenge, but we must take into account that it was only applied to one university, so it is limited, but the model can be used by applying it to other universities in the country and abroad, in our case in Turkey. The model as seen finds many elements in common with what other researchers have identified with reference to the behavior of students from different countries regarding plastic waste.

**Author Contributions:** Conceptualization, G.D.B. and S.S.; methodology, G.D.B. and S.S.; writing—original draft preparation, G.D.B. and S.S.; writing—review and editing, G.D.B. and S.S. All authors have read and agreed to the published version of the manuscript.

**Funding:** This research received no external funding.

**Institutional Review Board Statement:** The study was conducted in accordance with the Declaration of Helsinki, and approved by Ethics Committee of Technical University of Cluj Napoca (CEU 515/20/03/2023).

**Informed Consent Statement:** Informed consent was obtained from all subjects involved in the study.

**Conflicts of Interest:** The authors declare no conflict of interest.

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
