# Peer review of "Effects of Romanian Student’s Awareness and Needs Regarding Plastic Waste Management"

_sustainability, doi:10.3390/su15086811_

Round 1

Reviewer 1 Report

The paper sounds interesting. I believe that it can effectively contribute to the waste management practices in the plastic sector after a revision. In this regard, I have some comments to hopefully help the authors increase the quality of their presentation and the content of the manuscript. Please find my comments in the following.

1- Professional proofreading is required. I found many typos and grammatical errors in sentence structures.

2- For some statements, you need to cite their references. For instance, in the introduction section “In order to join Europe, a stand-alone place was occupied by Chapter 22 "Environmental protection", and within it the position "Waste" was noted”. You should refer to the relevant reference. The same comment for this sentence: “Germany and Italy are managing very well the management of recycling plastic in comparison with Romania and Finland which occupied the last place”. The same comment for this sentence: “Romania recycles only 13% of waste, and the rate of disposal of waste by final disposal at landfills is 69%, among the highest in Europe”. There are so many other statements that need reference support. This is not acceptable in academia and you must carefully check such issues when writing a paper. Please check this issue within the whole manuscript.

3- In the introduction section the authors have written “Our country has committed itself to…”, but they do not name any country. Please pay attention that you have submitted your article to an international journal, and you should be more careful with some amendments in this regard.

4- In the literature review section, there are many short paragraphs that should be linked with each other. Please try to incorporate some of the paragraphs to provide a better and more solid flow of information. Besides, you can refer to the significant effect of the pandemic on plastic waste generation and management. In this regard, you might refer to the paper titled “waste management beyond the covid-19 pandemic: bibliometric and text mining analyses”.

5- Please avoid reference lumping. For each statement, 3 references are adequate. Otherwise, you should separate the content. For instance, in line 285, you have cited [7], [10], [11], [17], [19-20], [22-23], [36-37], [45], all together.

6- The conclusion section should be thoroughly rewritten. The first sentence with many references should be removed and presented in a better way. No need to cite many references here since this is the conclusion of your research, not the other researchers'. In this section, the main finding of your research, its contribution to the domain, and its main implications for practice and theory should be highlighted. Besides, I did not understand why the authors presented section 6.1 “A SWOT Analyze for Romanian students” in the conclusion section. Why here??? Should you not present this above the conclusion? I recommend revisiting this section. In addition, the sections are too short, and I think there is no need to name the sub-sections. For instance, section 6.2 is only 2 sentences.

Author Response

Response to Reviewer 1 Comments

Dear Editor

Dear Reviewer

I very much appreciated the encouraging, critical and constructive comments on this manuscript by the reviewer. The comments have been very thorough and useful in improving the manuscript. I strongly believe that the comments and suggestions have increased the scientific value of revised manuscript by many folds. I have taken them fully into account in revision. I am submitting the corrected manuscript with the suggestion incorporated the manuscript. The manuscript has been revised as per the comments given by the reviewer, and our responses to all the comments are as follows:

The paper sounds interesting. I believe that it can effectively contribute to the waste management practices in the plastic sector after a revision. In this regard, I have some comments to hopefully help the authors increase the quality of their presentation and the content of the manuscript. Please find my comments in the following.

Point 1. Professional proofreading is required. I found many typos and grammatical errors in sentence structures.

Response 1: We made  the correction. The article will be verify by the professional personal from MDPI

Point 2- For some statements, you need to cite their references. For instance, in the introduction section “In order to join Europe, a stand-alone place was occupied by Chapter 22 "Environmental protection", and within it the position "Waste" was noted”. You should refer to the relevant reference.

Response 2: I made  the correction.

Sustainable development has a key component to environmental issues which is one of the Union's horizontal policies European.  In order to join Europe, a stand-alone place was occupied by Chapter 22 "Environmental protection", and within it the position "Waste" was noted [1].

  1. Plastic waste and recycling in the EU: facts and figures | NewsWaste recycling - European Environment Agency, Available online:

Point 3. The same comment for this sentence: “Germany and Italy are managing very well the management of recycling plastic in comparison with Romania and Finland which occupied the last place”.

Response 3: I delete the paragraph. Too many  details

Point 4. The same comment for this sentence: “Romania recycles only 13% of waste, and the rate of disposal of waste by final disposal at landfills is 69%, among the highest in Europe”.

Response 4: I made the correction

 Romania recycles only 13% of waste, and the rate of disposal of waste by final disposal at landfills is 69%, among the highest in Europe [5].

Point 5. There are so many other statements that need reference support. This is not acceptable in academia and you must carefully check such issues when writing a paper. Please check this issue within the whole manuscript.

Response 5: I made the correction in text and put to References.

Statistics showed that almost a third of plastic waste is recycled in Europe plastic production has grown exponentially worldwide in just a few decades, from 1.5 million tons in 1950 to 359 million tonnes in 2018 [2].

After a sharp decline in the first half of 2020 due to Covid-19, production plastic recovered in the second half of the year and with that came plastic waste [2] but EU is already taking steps to reduce the amount of plastic waste. Androniceanu et al.[3]  estimate that in 2019, globally, the production and incineration of plastics released more than 850 million tons of greenhouse gases into the atmosphere and by 2050, these emissions could reach 2.8 billion tons. Some of these can be avoided by better recycling.

Unfortunately, Romania is among the 14 EU member states that risk not being able to meet the 50% waste recycling target in 2020. The European Commission's analysis shows that the main causes of the problem are related to the fact that [4]: Although recycling cannot replace the need to significantly reduce the amount of disposable packaging and is by no means a justification for increasing plastic production, it has an important role to play in the transition to a plastic-free economy [6].

Bastos de Sousa [7] mentioned the essential role of the Agenda 2030 for Sustainable Development [8], which sets the Sustainable Development Goals and provides a critical overview of the role of plastic, mentioning its advantages and disadvantages.

Added to References

  1. 2. Plastic waste and recycling in the EU: facts and figures, Available online: https://www.europarl.europa.eu/news/ro/ headline

Point 5- In the introduction section the authors have written “Our country has committed itself to…”, but they do not name any country. Please pay attention that you have submitted your article to an international journal, and you should be more careful with some amendments in this regard.

Response 2: I made  the correction.

Romania has committed itself to a substantial reduction in a period of 15 years quantities of waste, to make the necessary systems for collection, recycling  and revaluation, but also those of future protection of the environment and a population against pollution.

Point 6- In the literature review section, there are many short paragraphs that should be linked with each other. Please try to incorporate some of the paragraphs to provide a better and more solid flow of information. Besides, you can refer to the significant effect of the pandemic on plastic waste generation and management. In this regard, you might refer to the paper titled “waste management beyond the covid-19 pandemic: bibliometric and text mining analyses”.

Response 6: I introduce in text   

A significant effect of the pandemic on plastic waste generation and management was register. Prasetiawan and Wasisto[24] investigated the relationship between students' perception and attitudes in waste management using the Internet leads to changes in attitude and behavior in society. Several studies have also highlighted the positive impact of COVID-19 on the environment, while limited published studies the long-term negative impact of COVID-19 on the environment and waste management. A negative aspect was observed by Ranjbari et al.[25] and Mohamed et al.[26] because during the COVID-19 pandemic there was a significant increase in the demand for personal protective equipment, face masks, increasing globally and the amount of plastic waste in the health sector. The effects of the COVID-19 pandemic were also felt on the production, use and waste of plastic materials, on economic activities and the use of plastic materials. The use of plastic materials decreased in the industrial sectors, as did the consumption of plastic materials in construction, the automotive industry decreased. The pandemic has led to an increased demand for single-use plastic, increasing the pressure on this already out-of-control problem. Peng et al.[27] noted that plastic waste also harms marine life and has become a major environmental concern worldwide.

Point 7- Please avoid reference lumping. For each statement, 3 references are adequate. Otherwise, you should separate the content. For instance, in line 285, you have cited [7], [10], [11], [17], [19-20], [22-23], [36-37], [45], all together.

Response 7: I made  the correction. New version

As a conclusion until now, the main focus of research has been on high school, college, university students around the world on the problem of waste plastic, on waste management, or recycling and reuse of plastic, and the attitude of students towards the new biodegradabile plastic [11], [24], [29], students’ behavior towards the environment [19], [46], students’ behavior towards plastic waste [20], [22], [24], students’ knowledge about plastic pollution [28], [42], [45]], students’ knowledge about the plastic waste [9], [11], [30], students’ awareness about plastic [19], [46] and students’ needs [44], [47]. Environmental protection was raised to the rank of objective of major public interest and responsibility, based on the fundamental principles of environmental law: the principle of conservation, the principle of prevention, the principle of precaution in decision-making and the polluter pays. Taking in consideration that literature has so far focused on specific aspects of plastic waste and sustainability in the higher education sector, the aim of this study is to explore the impact toward of Romanian students’.

Point 8- The conclusion section should be thoroughly rewritten.

No need to cite many references here since this is the conclusion of your research, not the other researchers'.

In this section, the main finding of your research, its contribution to the domain, and its main implications for practice and theory should be highlighted

Response 8: I made  the correction. Rewritten Conclusion

The results obtained after filling out the questionnaire led to the creation of the new model and identifying common elements of students’ needs, students’ behavior a students’awareness regarding waste plastic. For our study Romanian students’ awareness obtain the highets value so they are involve in waste plastic management. We also verified, the influence of the profile to which the students are enrolled in our case the management profile students knowledge are higher from theoretical point of view but engineering students are more practical, and they are involved in activities and active participation organized by university or they are attend conferences on plastic topic.  Authors identify the similarities and differences when we take in consideration students’ gender, male are more practical in cleaning  the nature from plastic, in comparison with female students which pay attention to small details of using, selecting the waste plastic. Students’ are acting individual as person and they don’t like to be involve in extra activities. They are inform from mass media, from university but sometimes they are lazy to follow the rules or they consider that are qualify person who are payed to do their job. The results are a signal for universities which must to involve students in activities with specific topic about environment protection and waste plastic, in volunteer activities and also in research activities to understand the importance of subject. In Romania, efforts to combat the plastic pollution crisis are rather minor, fragmented, and generally ignore the extremely harmful consequences that both  we and the environment have to bear every day. Also on the study of bio plastic, and the future vision without plastics present also that the respondents and also the organization must to be careful when it comes to alternative plastic (cane or corn starch product, biodegradable and / or compostabel and analyze all the sustainability issues associated with this type of plastic. For the compact segment identified in the research study of 20-30 % percent of the respondents who have no opinion and do not want to be involved in any kind of activities just sit, we can specify the following factors: ·           the lack of concern for this problem, first of all by the shops and then by authorities who do not properly inform about the problem of plastic;

The Cross Model for students in relation to bio plastic obtained a low value, which means a weak connection between the needs of students that influence their behavior. Connection needs present a weak point in the education of students regarding environmental protection with plastic issues as a direct target. So a necessity for students’ needs, in their behavior towards sustainable environment education is influence by knowledge. 

Precisely , the factors that led to this disastrous situation are the following:

·           lack of a separate collection system at source (at people's homes) on at least 4-5 types of waste: paper / cardboard, plastic / metal, glass , bio waste (food and vegetable waste) and mixed; and total lack of waste collection in rural areas, where there are no sanitation contracts, which leads most often either when disposing of them in the wild or when burning uncontrolled.The strongest link is the one between the students' awareness influences their information needs regarding the plastic problem. This is where the role of universities comes into play by: ·           creating platforms for courses specific to the field, creating virtual libraries as a source of useful information;·           the transfer of information between different universities or student groups for common research topics.The students' awareness influences their behavior by getting involved in voluntary actions, through the individual actions of selecting plastic at home or at university, protecting nature by choosing biodegradable products when shopping.

Point 9. Besides, I did not understand why the authors presented section

6.1 “A SWOT Analyze for Romanian students” in the conclusion section. Why here??? Should you not present this above the conclusion?

I recommend revisiting this section. In addition, the sections are too short, and I think there is no need to name the sub-sections. For instance, section 6.2 is only 2 sentences.

 Response 9: I put the SWOT Analyze in Discussion section.

Response to Reviewer 1 Comments

Dear Editor

Dear Reviewer

I very much appreciated the encouraging, critical and constructive comments on this manuscript by the reviewer. The comments have been very thorough and useful in improving the manuscript. I strongly believe that the comments and suggestions have increased the scientific value of revised manuscript by many folds. I have taken them fully into account in revision. I am submitting the corrected manuscript with the suggestion incorporated the manuscript. The manuscript has been revised as per the comments given by the reviewer, and our responses to all the comments are as follows:

The paper sounds interesting. I believe that it can effectively contribute to the waste management practices in the plastic sector after a revision. In this regard, I have some comments to hopefully help the authors increase the quality of their presentation and the content of the manuscript. Please find my comments in the following.

Point 1. Professional proofreading is required. I found many typos and grammatical errors in sentence structures.

Response 1: We made  the correction. The article will be verify by the professional personal from MDPI

Point 2- For some statements, you need to cite their references. For instance, in the introduction section “In order to join Europe, a stand-alone place was occupied by Chapter 22 "Environmental protection", and within it the position "Waste" was noted”. You should refer to the relevant reference.

Response 2: I made  the correction.

Sustainable development has a key component to environmental issues which is one of the Union's horizontal policies European.  In order to join Europe, a stand-alone place was occupied by Chapter 22 "Environmental protection", and within it the position "Waste" was noted [1].

  1. Plastic waste and recycling in the EU: facts and figures | NewsWaste recycling - European Environment Agency, Available online:

Point 3. The same comment for this sentence: “Germany and Italy are managing very well the management of recycling plastic in comparison with Romania and Finland which occupied the last place”.

Response 3: I delete the paragraph. Too many  details

Point 4. The same comment for this sentence: “Romania recycles only 13% of waste, and the rate of disposal of waste by final disposal at landfills is 69%, among the highest in Europe”.

Response 4: I made the correction

 Romania recycles only 13% of waste, and the rate of disposal of waste by final disposal at landfills is 69%, among the highest in Europe [5].

3.          S

Point 5. There are so many other statements that need reference support. This is not acceptable in academia and you must carefully check such issues when writing a paper. Please check this issue within the whole manuscript.

Response 5: I made the correction in text and put to References.

Statistics showed that almost a third of plastic waste is recycled in Europe plastic production has grown exponentially worldwide in just a few decades, from 1.5 million tons in 1950 to 359 million tonnes in 2018 [2].

After a sharp decline in the first half of 2020 due to Covid-19, production plastic recovered in the second half of the year and with that came plastic waste [2] but EU is already taking steps to reduce the amount of plastic waste. Androniceanu et al.[3]  estimate that in 2019, globally, the production and incineration of plastics released more than 850 million tons of greenhouse gases into the atmosphere and by 2050, these emissions could reach 2.8 billion tons. Some of these can be avoided by better recycling.

Unfortunately, Romania is among the 14 EU member states that risk not being able to meet the 50% waste recycling target in 2020. The European Commission's analysis shows that the main causes of the problem are related to the fact that [4]: Although recycling cannot replace the need to significantly reduce the amount of disposable packaging and is by no means a justification for increasing plastic production, it has an important role to play in the transition to a plastic-free economy [6].

Bastos de Sousa [7] mentioned the essential role of the Agenda 2030 for Sustainable Development [8], which sets the Sustainable Development Goals and provides a critical overview of the role of plastic, mentioning its advantages and disadvantages.

Added to References

  1. 2. Plastic waste and recycling in the EU: facts and figures, Available online: https://www.europarl.europa.eu/news/ro/ headline

Point 5- In the introduction section the authors have written “Our country has committed itself to…”, but they do not name any country. Please pay attention that you have submitted your article to an international journal, and you should be more careful with some amendments in this regard.

Response 2: I made  the correction.

Romania has committed itself to a substantial reduction in a period of 15 years quantities of waste, to make the necessary systems for collection, recycling  and revaluation, but also those of future protection of the environment and a population against pollution.

Point 6- In the literature review section, there are many short paragraphs that should be linked with each other. Please try to incorporate some of the paragraphs to provide a better and more solid flow of information. Besides, you can refer to the significant effect of the pandemic on plastic waste generation and management. In this regard, you might refer to the paper titled “waste management beyond the covid-19 pandemic: bibliometric and text mining analyses”.

Response 6: I introduce in text   

A significant effect of the pandemic on plastic waste generation and management was register. Prasetiawan and Wasisto[24] investigated the relationship between students' perception and attitudes in waste management using the Internet leads to changes in attitude and behavior in society. Several studies have also highlighted the positive impact of COVID-19 on the environment, while limited published studies the long-term negative impact of COVID-19 on the environment and waste management. A negative aspect was observed by Ranjbari et al.[25] and Mohamed et al.[26] because during the COVID-19 pandemic there was a significant increase in the demand for personal protective equipment, face masks, increasing globally and the amount of plastic waste in the health sector. The effects of the COVID-19 pandemic were also felt on the production, use and waste of plastic materials, on economic activities and the use of plastic materials. The use of plastic materials decreased in the industrial sectors, as did the consumption of plastic materials in construction, the automotive industry decreased. The pandemic has led to an increased demand for single-use plastic, increasing the pressure on this already out-of-control problem. Peng et al.[27] noted that plastic waste also harms marine life and has become a major environmental concern worldwide.

Point 7- Please avoid reference lumping. For each statement, 3 references are adequate. Otherwise, you should separate the content. For instance, in line 285, you have cited [7], [10], [11], [17], [19-20], [22-23], [36-37], [45], all together.

Response 7: I made  the correction. New version

As a conclusion until now, the main focus of research has been on high school, college, university students around the world on the problem of waste plastic, on waste management, or recycling and reuse of plastic, and the attitude of students towards the new biodegradabile plastic [11], [24], [29], students’ behavior towards the environment [19], [46], students’ behavior towards plastic waste [20], [22], [24], students’ knowledge about plastic pollution [28], [42], [45]], students’ knowledge about the plastic waste [9], [11], [30], students’ awareness about plastic [19], [46] and students’ needs [44], [47]. Environmental protection was raised to the rank of objective of major public interest and responsibility, based on the fundamental principles of environmental law: the principle of conservation, the principle of prevention, the principle of precaution in decision-making and the polluter pays. Taking in consideration that literature has so far focused on specific aspects of plastic waste and sustainability in the higher education sector, the aim of this study is to explore the impact toward of Romanian students’.

Point 8- The conclusion section should be thoroughly rewritten.

No need to cite many references here since this is the conclusion of your research, not the other researchers'.

In this section, the main finding of your research, its contribution to the domain, and its main implications for practice and theory should be highlighted

Response 8: I made  the correction. Rewritten Conclusion

The results obtained after filling out the questionnaire led to the creation of the new model and identifing common elements of students’ needs, students’ behavior a students’awareness regarding waste plastic. For our study Romanian students’ awareness obtain the highes value so they are involve in waste plastic management. We also verified, the influence of the profile to which the students are enrolled in our case the management profile students knowledges are hight from theoretical point of view but engineerings students are more practical, and they are involved in activites and active participation organized by universityor they are attenting conferences on plastic topic.  Authors identify the similarities and differences when we take in consideration students’ gender, male are more practical in cleaning  the nature from plastic, in comparison with female students which pay attention to small details of using, selecting the waste plastic. Students’ are acting individual as person and they don’t like to be involve in extra activities. They are inform from mass media, from university but sometimes they are lazy to follow the rules or they consider that are qualify person who are payed to do their job. The results are a signal for universites which must to involve students in activities with specific topic about environment protection and waste plastic, in volunteer activities and also in research activities to understand the importance of subject. In Romania, efforts to combat the plastic pollution crisis are rather minor, fragmented, and generally ignore the extremely harmful consequences that both  we and the environment have to bear every day. Also on the study of bio plastic, and the future vision without plastics present also that the respondents and also the organization must to be careful when it comes to alternative plastic (cane or corn starch product, biodegradable and / or compostable) and analyze all the sustainability issues associated with this type of plastic. For the compact segment identified in the research study of 20-30 % percent of the respondents who have no opinion and do not want to be involved in any kind of activities just sit, we can specify the following factors: ·           the lack of concern for this problem, first of all by the shops and then by authorities who do not properly inform about the problem of plastic;

The Cross Model for students in relation to bio plastic obtained a low value, which means a weak connection between the needs of students that influence their behavior. Connection needs present a weak point in the education of students regarding environmental protection with plastic issues as a direct target. So a necessity for students’ needs, in their behavior towards sustainable environment education is influence by knowledge. 

Precisely , the factors that led to this disastrous situation are the following:

·           lack of a separate collection system at source (at people's homes) on at least 4-5 types of waste: paper / cardboard, plastic / metal, glass , bio waste (food and vegetable waste) and mixed; and total lack of waste collection in rural areas, where there are no sanitation contracts, which leads most often either when disposing of them in the wild or when burning uncontrolled.The strongest link is the one between the students' awareness influences their information needs regarding the plastic problem. This is where the role of universities comes into play by: ·           creating platforms for courses specific to the field, creating virtual libraries as a source of useful information;·           the transfer of information between different universities or student groups for common research topics.The students' awareness influences their behavior by getting involved in voluntary actions, through the individual actions of selecting plastic at home or at university, protecting nature by choosing biodegradable products when shopping.

Point 9. Besides, I did not understand why the authors presented section

6.1 “A SWOT Analyze for Romanian students” in the conclusion section. Why here??? Should you not present this above the conclusion?

I recommend revisiting this section. In addition, the sections are too short, and I think there is no need to name the sub-sections. For instance, section 6.2 is only 2 sentences.

 Response 9: I put the SWOT Analyze in Discussion section.

Reviewer 2 Report

The subject is within the scope of the Journal and Special Issue. The article has weak points that must be improved. Some suggestions to improve the study:  

- The title of the paper should be more specific in terms of the content.

The problem and the main results do not sound clear enough in the abstract, please revised.  

- Research gap and RQs are missing in the Intro section, please clarify.

- For clearer understanding of the steps and methods of your study it would be better to provide Research methodology section. Please revise accordingly.

- The conclusion section is too length and does not reveal the main results of the study and their practical implication. Discussion section can be added.

- The academic writing of the article should to be improved.

- The literature is presented in gross violation of the requirements of the journal.

Author Response

Response to Reviewer 2 Comments

Dear Editor 
Dear Reviewer 
I very much appreciated the encouraging, critical and constructive comments on this manuscript by the reviewer. The comments have been very thorough and useful in improving the manuscript. I strongly believe that the comments and suggestions have increased the scientific value of revised manuscript by many folds. I have taken them fully into account in revision. I am submitting the corrected manuscript with the suggestion incorporated the manuscript. The manuscript has been revised as per the comments given by the reviewer, and our responses to all the comments are as follows: 

The subject is within the scope of the Journal and Special Issue. The article has weak points that must be improved. Some suggestions to improve the study:  
Point 1 The title of the paper should be more specific in terms of the content.
Response 1.  We change the title 
Effects of Romanian student’s awareness and needs regarding waste plastic management 
Point 2. The problem and the main results do not sound clear enough in the abstract, please revised. ‘
 Response 2.  I made the suggested correction

Abstract:. Abstract:. The study was applied to a group of 537 students from the University of Cluj Napoca, Romania, respectively from the engineering and management specializations. An online questionnaire was applied using 29 questions and was structure in four parts. The first part to identify the students’ characteristics (gender, field of specialization, participation and attending in activities in the field and if he/she is an environmentalist). Second part  to determine the awareness of students’ about plastic and plastic pollution. Another part to determine the needs of students and the ways of learning and information. The last part allows determining the behavior of students in their daily life ( use of bio plastic bags, environmental protection). The results show that students have enough information about bio degradable plastic but they act depending on the situation, respecting or not the rules for selecting plastic waste. The female student’ are paying a lot of attention in selecting and choosing bio plastic products.The male students’ are direct involved in cleaning the green nature. Management students’ pay attention to small details in compared to engineering students who choose bio plastic even though the costs are higher. The model show that strongest connection is between students awareness about the plastic problem and the need to adapt to new regulations. Their awareness regarding the waste plastic depend of their needs and behavior. Also their needs influence their behavior.  Using the model universities can promote the importance of bio plastic through study programs or by involving students in volunteer activities, through their active involvement in environmental protection and selective waste recycling..
Point 3.  Research gap and RQs are missing in the Intro section, please clarify.
Response 3. 
5. Results 
5.1. Students characteristics
The analysis carried out from the point of view of the students, whatever they are ecologists in 36.69% percent or not in 63.31% percent, gave the following results, taking into account the gender like in Table3. It is very important also to mention that only 24.58% percent from students’ attend a conference on nature conservation or about plastic problem (Table 3). Conversely a majority of 75.42 % percent of students did not attend any conference or any activity about the protection off the environment and nature regarding plastic waste boys is more involved in a percentage of 13.22%. Also they mention in a 71.88% that they never take part to environmental activities organized within university and only 28.1 % participate in such activities organized by university from which again boys are more active in a percentage of 18,06%. 36.69 % of students consider themselves environmentalists, of which 17.32% are girls and a majority of 53.45% consider themselves environmentalists only sometimes depending on the situation.
In conclusion, young students are comfortable and do not get involved in extracurricular activities because they spend their free time with other activities.  

Table 3. Students individual characteristics regarding active participation
Gender    F    M    Total
Have you attended a conference on nature conservation before?    Yes    61    71    132
    No    172    233    405
Total    233    304    537
Do you think you are an environmentalist?    Yes    93    104    197
    No    25    28    53
    Sometimes    115    172    287
Total    233    304    537
Did you take part in environmental activities organized within 
the university?    Yes    54    97    151
    No    178    207    386
Total    233    304    537

Table 4 present the correlation between student’s specialization and their individual characteristics. From 537 respondents, 53.63 % are engineering students from IT, automatization  and electrical specialization and 46.37% from management and business administration. The management students’ admit in 38.15 % percent that they are ecologists and only sometimes 51.81 % when they remember or are imposed rules that they must follow. 

Table 4. Students participation in function of specialization 
    In which department do you study?    Total
                    Automatization          IT     Management / Business    Electrical    
Did you take part in environmental activities organized within the university?    Yes    48    42    20    41    151
    No    66    47    229    45    386
Total    114    89    249    85    537
Have you attended a conference on nature conservation before?    Yes    26    25    50    31    132
    No    88    64    199    54    405
Total    114    89    249    85    537
I  know that petroleum product plastics take a long time to biodegrade?    Yes    96    79    219    67    461
    No    7    4    10    6    27
    No idea    11    6    20    12    49
Total    114    89    249    85    537
Do you think you are an environmentalist?    Yes    48    24    95    30    197
    No    9    13    25    6    53
    Sometimes    57    52    129    49    287
     Total    114    89    249    85    537

85.85% of the students from both specializations have information related to biodegradable plastic and the benefits of its use. A majority of 87.95% was obtained by the management students’. Although theoretically they are better informed than those from engineering, in practice they do not participate in the environmental activities organized by the university in a massive percentage of 91.97%. When asked about participation in conferences about nature conservation, a percentage of 79.91% of them stated that they do not participate, citing that their profile refers to the economic side of bio plastic.
We can conclude that students are not involved in collateral activities with their field of study, but as individuals some of them apply the rules of where plastic waste comes from. Students in both courses agree with the rules and special selection of plastic waste and prefer to use low-degradation items. Boys are more involved in practical activities related to nature, or participation in actions organized by the university. The girls have the information about bio plastic but prefer not to be actively involved. 
So the question that arises is how to attract the younger generation to be more active and involved in this issue.
 In order to realize the cross model and find solutions, we also studied the correlation  between students' awareness and behavior in function of their specialization and gender to obtain  more  information. 
5.4. Students Needs
Teddy Prasetiawan and Wasisto [24] sustain that education about waste in schools plays a significant role in increasing students' perceptions of the environment. Previous research has suggested that students' perceptions in waste management plastic can positively impact future generations' preparation as agents that drive changes in attitude and behavior in society. Table 6 present the results obtain for students’ needs items N1-N5. 

            Table 6. Items for students’ needs regarding bio plastic 
N1     I would like to learn new information about the use of bioplastic products.
N2     Bioplastics should replace conventional polymers in the future.
N3     I think that all kinds of studies on the pollution of traditional plastic products should be increased.
N4     I think that bioplastics should be used in mass social events 
N5     I think that bioplastics should be used in takeaway products.

Maybe the 43.4 % from students’ are aware that this process requires a long time and tailor-made strategies.
56.6 % percent from students don’t want to learn new information about bioplastic, maybe their specialeties are different and they consider because their specialty is different and they consider that what they know is enough or the information they have through mass media promotion is enough. In a percent of 13.2% some students’ are still aware of the importance of plastic waste and consider it’s necessary to continue research in this field of removing plastic and replacing it in time. 
For item N1:‘’I would like to learn new information about the use of bioplastic products.’’ only 20.3 % percent from students from both specialeties are open to learn and to obtain new information about bio plastic , maybe because some of them are already working, or they don’t want to spend extra time with new information, mention are qualify person who are already payed to do specific job. 
For item N2: ‘’Bioplastics should replace conventional polymers in the future ‘’ a 30.5% percent from students’are agree and totally agree that in future plastic must to be replace with the new material biodegradable, to prevent the pollution. 
For item N5:‘’I think that bioplastics should be used in takeaway products.’’ The biggest value of 31.5 % percent of students agree with the bioplastic takeaway maybe because they know the importance to protect the environment and they have an environmental education already. 
But for item N4:‘’I think that bioplastics should be used in mass social events’’ the percent it’s increasing to 31.1 % so students  know about the waste plastic  esspecialy from social events where they are involved. 
In conclusion students have information, knowledge about plastic and waste plastic, they already apply and are oriented in their daily life  to  use the new material bioplastic, but they don’t want to be involved in courses or lectures on that specific topic in their extra curricula time.
5.5. Students’Awareness
 From awareness point of view the results for items A1-A8 from Table 5, present that students are inform and they are concerned about the issue of plastic waste and plastic pollution, but they are acting individual. 
For item A4: ‘’ I prefer the products obtained from the bioplastics industry because they are renewable’’ the biggest value of  33.9 % percent form students from both specialization management and engineering prefer bioplastic products because they are renewable, that;s mean they are inform about the waste plastic problem. 
For item A5: ‘’ I prefer to avoid products that increase global warming.’’ 32% percent from students’ agree that plastic waste problem can increase the global warning.
For item A6 ’I prefer bioplastic products because they do not harm the nature when they decompose’’ the biggest value of  33% percent from respondents agree with bioplastic because the material don’t harm the nature when they decompose.

                Table 5.  Awarness Items selection
A1     I prefer products that are less harmful to nature
A2     I use low-damage objects such as paper and glass.
A3     I recognize the bioplastic logos on the products.
A4     I prefer the products obtained from the bioplastics industry because they are renewable.
A5     I prefer to avoid products that increase global warming.
A6     I prefer bioplastic products because they do not harm the nature when they decompose.
A7     I prefer bioplastic products as they do not harm human health when degraded.
A8     I prefer bioplastic products because they degrade earlier in nature.

For item A7: ‘’I prefer bioplastic products as they do not harm human health when degraded’’ a 30.1% percent from students’ are agree and totally agree to choose the bioplastic products which are not harm health after the degraded. 
For item A8: ‘’ I prefer bioplastic products because they degrade earlier in nature’’ a value of  32,1% percent from respondents have information about the bioplastic degradation, in time and they are awareness about the environment protection, a percent of 26.1% they don’t have any opinion, they are passive.
For all items a compact value between 20 - 30% from respondents are staying and watching and consider that there are secialist payed to take care of  plastic  problem.   
5.6. Students Behavior
Students’ behavior from gender point of view,  behaviors were shown in purchasing products with plastic packaging, preparing shopping bag, re-using plastic bags, taking own meal box, and having food on the sites to reduce single used plastic package. Students’ as target it is important as these individuals will be the ones making decisions priestly and politically, with regards to recycling programs in their communities. 
By understanding their behavior in colleague and universities it is possible for those who design recycling program to develop and modify program in function if their needs and information program for their education. In Table 7 we present the students’ behavior items survey B1-B9.
For item B2:’’ When I see plastic being burned, I warn it, thinking it will cause air pollution.’’ a percent of 31.3 % of students have a positive behavior when they are involve in different plastic waste situation which can harm the nature and polluted.  

                           Table 7. Items for students’behavior regarding waste plastic
B1    I buy bioplastic products, even if they are expensive.
B2    When I see plastic being burned, I warn it, thinking it will cause air pollution.
B3    I use bioplastic bags for my grocery shopping.
B4    I make an effort to use a small number of bags in my daily life.
B5    I do not throw plastic products into nature.
B6    I throw the recycling products into the relevant boxes.
B7    I use mesh/cloth/paper bags instead of using disposable bags while shopping.
B8    I inform the people around me about the use of bioplastic products.
B9    When I see plastic in the green area, I take it from there.

    For item B7:’’I use mesh/cloth/paper bags instead of using disposable bags while shopping.’’ a 30. 4 % percent are trying to adapt their behavior with the new standards, the low value obtained it is maybe because they are ot prepare yet with new situation or maybe the cost are increasing a little bit.  
For item B8: ‘’I inform the people around me about the use of bioplastic products.’’ 33 %  percent of students’ from both fields agree with the importance of dissemination and information regarding the use of bioplastic products.
For item B9: ‘’When I see plastic in the green area, I take it from there’’ the highest value of 33% percent of the students have a protective behavior towards nature and green nature by selecting waste in specific places.

Point 4- For clearer understanding of the steps and methods of your study it would be better to provide Research methodology section. Please revise accordingly.
Response 4. 
4. Research Methodology 
4.1. Materials and Methods
The purpose of the study was to understand the students' behavior and awareness regarding the sustainable environment and students’ behavior and knowledge with reference to the phenomenon of plastic recycling. A total of 537 students were involved in an experimental study to identify information and the level of knowledge with reference to the current problem of plastic and the need to replace it with bio plastic. 
The questionnaire was applied online between June and July 2022 at the Technical University of Cluj Napoca, Romania, from department of economics and engineering. The survey was structure in three parts: 
Part 1. Students’ characteristics (gender, faculty, specialization, participation and attending activities about environment protection and their environmental attitude (I1-I7 questions );
Part 2: Students’ awareness (A1-A8 questions) to examine whether the young generation plays a responsible role towards plastic products and their perception towards this topic plastic and new biodegradable plastic;
Part 3. Students’ needs (N1-N5 questions) perception of the concept of plastic sustainability and their participation in different activities regarding plastic recycling
Part 4. Students' behavior (B1-B9 questions) about healthy plastic education,  how many of them select plastic, if they use ecological products. 
In this study, we’re used Google Drive form to create the survey, and apply online, data analyze and statistical processing were performed using the SPSS software package and for students’ model Rigle et al.[59], Starstedt et al.[60] and Hair et al.[61] SmartPLS 3 program.

4.2.    Sample and Measurement Tool
To measure the student’s behavior, needs and awareness on waste plastic, a Likert-scale type questionnaire, ranging from 1 ‘Totally Appropriate’ to 5 ‘Not at all appropriate’, was applied. Research model is given in Figure 1 we’re taken into consideration: N- Needs; A-awareness about environment and plastic and B-Behavior. 

Figure 1. Research model regarding plastic waste. Source: By authors

In this study, the following factors were taken in consideration: 
 B- students behavior (B1, B2, B3, B4, B5, B6, B7, B8 and B9 questions like in Table 4). 
A- students awareness regarding waste plastic ( A1, A2, A3, A4, A5, A6, A7 and A8 questions like in Table 5) ;
N- students needs ( N1, N2, N3, N4 and N5 questions like in Table 6);
To determine the dimensions of students’ behavior and identify students’ needs and awareness about waste plastic and bioplastic an Explanatory Factor Analysis (EFA) were applied to the data set. By independent samples t test, the hypotheses were tested using the SPSS statistical analyses software.
Hypothesis for the model given in  Figure 1 are as below:
H1: Students Awareness towards plastic waste effect their Needs;
H2: Students Awareness towards plastic waste effect their Behavior;
H3: Students Needs toward plastic waste effect their Behavior.

Point 5. The conclusion section is too length and does not reveal the main results of the study and their practical implication. 
Response 5. I made  the correction. Rewritten Conclusion
The results obtained after filling out the questionnaire led to the creation of the new model and identifying common elements of students’ needs, students’ behavior and students’awareness regarding waste plastic. 
For our study Romanian students’ awareness obtain the highes value so they are involve in waste plastic management. 
We also verified, the influence of the profile to which the students are enrolled in our case the management profile students knowledge are hight from theoretical point of view but engineering students are more practical, and they are involved in activities and active participation organized by university or they are attending conferences on plastic topic.  
Authors identify the similarities and differences when we take in consideration students’ gender, male are more practical in cleaning  the nature from plastic, in comparison with female students which pay attention to small details of using, selecting the waste plastic. 
Students’ are acting individual as person and they don’t like to be involve in extra activities. They are inform from mass media, from university but sometimes they are lazy to follow the rules or they consider that are qualify person who are payed to do their job. The results are a signal for universities which must to involve students in activities with specific topic about environment protection and waste plastic, in volunteer activities and also in research activities to understand the importance of subject. In Romania, efforts to combat the plastic pollution crisis are rather minor, fragmented, and generally ignore the extremely harmful consequences that both  we and the environment have to bear every day. 
Also on the study of bio plastic, and the future vision without plastics present also that the respondents and also the organization must to be careful when it comes to alternative plastic (cane or corn starch product, biodegradable and / or compostable) and analyze all the sustainability issues associated with this type of plastic. 
For the compact segment identified in the research study of 20-30 % percent of the respondents who have no opinion and do not want to be involved in any kind of activities just sit, we can specify the following factors: 
•    the lack of concern for this problem, first of all by the shops and then by authorities who do not properly inform about the problem of plastic;
•    lack of information is another reason specified by the respondents for the lack of information time .
The Cross Model for students in relation to bio plastic obtained a low value, which means  a weak connection between the needs of students that influence their behavior. Connection needs present a weak point in the education of students regarding environmental protection with plastic issues as a direct target. So a necessity for students’ needs, in their behavior towards sustainable environment education is influence by knowledge.  
Precisely , the factors that led to this disastrous situation are the following: 
•    lack of a separate collection system at source (at people's homes) on at least 4-5 types of waste: paper / cardboard, plastic / metal, glass , bio waste (food and vegetable waste) and mixed; 
•    lack of proper education among the population and decision-makers at central and local level;
•    lack of involvement of mayors and failure to assume their responsibilities;
•    the lack of sanctions and total lack of waste collection in rural areas, where there are no sanitation contracts, which leads most often either when disposing of them in the wild or when burning uncontrolled.
The strongest link is the one between the students' awareness influences their information needs regarding the plastic problem. This is where the role of universities comes into play by: 
•    creating platforms for courses specific to the field, creating virtual libraries as a source of useful information;
•    the transfer of information between different universities or student groups for common research topics.
The students' awareness influences their behavior by getting involved in voluntary actions, through the individual actions of selecting plastic at home or at university, protecting nature by choosing biodegradable products when shopping.

Point 6 .Discussion section can be added.
Response 6. I added 
Point 7. The academic writing of the article should to be improved.
Response 7.   We ask MDPI office professional service to verify English.
Point 8.- The literature is presented in gross violation of the requirements of the journal.
Response 8..  I made the correction.

Reviewer 3 Report

Need to revamp the whole thesis structure. Too amateur and please submit for proofreading. The findings were fine and good but very difficult to understand the coherence and the content. 

Author Response

Response to Reviewer 3 Comments

Dear Editor 
Dear Reviewer 
I very much appreciated the encouraging, critical and constructive comments on this manuscript by the reviewer. The comments have been very thorough and useful in improving the manuscript. I strongly believe that the comments and suggestions have increased the scientific value of revised manuscript by many folds. I have taken them fully into account in revision. I am submitting the corrected manuscript with the suggestion incorporated the manuscript. The manuscript has been revised as per the comments given by the reviewer, and our responses to all the comments are as follows: 

Point 1. Need to revamp the whole thesis structure. 
Response 1.  We change title 
Effects of Romanian student’s awareness and needs regarding waste plastic management 
Point 2. Too amateur and please submit for proofreading. 
Response  2.  We made the correction  
Point 3. The findings were fine and good but very difficult to understand the coherence and the content. 
Response  3.  We made correction for a better understanding 
Point 1 The title of the paper should be more specific in terms of the content.
Response 1.  
Point 2. The problem and the main results do not sound clear enough in the abstract, please revised. ‘
 Response 2.  I made the suggested correction

Abstract:. Abstract:. The study was applied to a group of 537 students from the University of Cluj Napoca, Romania, respectively from the engineering and management specializations. An online questionnaire was applied using 29 questions and was structure in four parts. The first part to identify the students’ characteristics (gender, field of specialization, participation and attending in activities in the field and if he/she is an environmentalist). Second part  to determine the awareness of students’ about plastic and plastic pollution. Another part to determine the needs of students and the ways of learning and information. The last part allows determining the behavior of students in their daily life ( use of bio plastic bags, environmental protection). The results show that students have enough information about bio degradable plastic but they act depending on the situation, respecting or not the rules for selecting plastic waste. The female student’ are paying a lot of attention in selecting and choosing bio plastic products.The male students’ are direct involved in cleaning the green nature. Management students’ pay attention to small details in compared to engineering students who choose bio plastic even though the costs are higher. The model show that strongest connection is between students awareness about the plastic problem and the need to adapt to new regulations. Their awareness regarding the waste plastic depend of their needs and behavior. Also their needs influence their behavior.  Using the model universities can promote the importance of bio plastic through study programs or by involving students in volunteer activities, through their active involvement in environmental protection and selective waste recycling..
Point 3.  Research gap and RQs are missing in the Intro section, please clarify.
Response 3.                            5. Results 
5.1. Students characteristics
The analysis carried out from the point of view of the students, whatever they are ecologists in 36.69% percent or not in 63.31% percent, gave the following results, taking into account the gender like in Table3. It is very important also to mention that only 24.58% percent from students’ attend a conference on nature conservation or about plastic problem (Table 3). Conversely a majority of 75.42 % percent of students did not attend any conference or any activity about the protection off the environment and nature regarding plastic waste boys is more involved in a percentage of 13.22%. Also they mention in a 71.88% that they never take part to environmental activities organized within university and only 28.1 % participate in such activities organized by university from which again boys are more active in a percentage of 18,06%. 36.69 % of students consider themselves environmentalists, of which 17.32% are girls and a majority of 53.45% consider themselves environmentalists only sometimes depending on the situation.
In conclusion, young students are comfortable and do not get involved in extracurricular activities because they spend their free time with other activities.  

Table 3. Students individual characteristics regarding active participation
Gender    F    M    Total
Have you attended a conference on nature conservation before?    Yes    61    71    132
    No    172    233    405
Total    233    304    537
Do you think you are an environmentalist?    Yes    93    104    197
    No    25    28    53
    Sometimes    115    172    287
Total    233    304    537
Did you take part in environmental activities organized within 
the university?    Yes    54    97    151
    No    178    207    386
Total    233    304    537

Table 4 present the correlation between student’s specialization and their individual characteristics. From 537 respondents, 53.63 % are engineering students from IT, automatizations and electrical specialization and 46.37% from management and business administration. The management students’ admit in 38.15 % percent that they are ecologists and only sometimes 51.81 % when they remember or are imposed rules that they must follow. 

Table 4. Students participation in function of specialization 
    In which department do you study?    Total
                    Automatization          IT     Management / Business    Electrical    
Did you take part in environmental activities organized within the university?    Yes    48    42    20    41    151
    No    66    47    229    45    386
Total    114    89    249    85    537
Have you attended a conference on nature conservation before?    Yes    26    25    50    31    132
    No    88    64    199    54    405
Total    114    89    249    85    537
I  know that petroleum product plastics take a long time to biodegrade?    Yes    96    79    219    67    461
    No    7    4    10    6    27
    No idea    11    6    20    12    49
Total    114    89    249    85    537
Do you think you are an environmentalist?    Yes    48    24    95    30    197
    No    9    13    25    6    53
    Sometimes    57    52    129    49    287
     Total    114    89    249    85    537

85.85% of the students from both specializations have information related to biodegradable plastic and the benefits of its use. A majority of 87.95% was obtained by the management students’. Although theoretically they are better informed than those from engineering, in practice they do not participate in the environmental activities organized by the university in a massive percentage of 91.97%. When asked about participation in conferences about nature conservation, a percentage of 79.91% of them stated that they do not participate, citing that their profile refers to the economic side of bio plastic.
We can conclude that students are not involved in collateral activities with their field of study, but as individuals some of them apply the rules of where plastic waste comes from. Students in both courses agree with the rules and special selection of plastic waste and prefer to use low-degradation items. Boys are more involved in practical activities related to nature, or participation in actions organized by the university. The girls have the information about bio plastic but prefer not to be actively involved. 
So the question that arises is how to attract the younger generation to be more active and involved in this issue.
 In order to realize the cross model and find solutions, we also studied the correlation  between students' awareness and behavior in function of their specialization and gender to obtain  more  information. 
5.4. Students Needs
Teddy Prasetiawan and Wasisto [24] sustain that education about waste in schools plays a significant role in increasing students' perceptions of the environment. Previous research has suggested that students' perceptions in waste management plastic can positively impact future generations' preparation as agents that drive changes in attitude and behavior in society. Table 6 present the results obtain for students’ needs items N1-N5. 

            Table 6. Items for students’ needs regarding bio plastic 
N1     I would like to learn new information about the use of bioplastic products.
N2     Bioplastics should replace conventional polymers in the future.
N3     I think that all kinds of studies on the pollution of traditional plastic products should be increased.
N4     I think that bioplastics should be used in mass social events 
N5     I think that bioplastics should be used in takeaway products.

Maybe the 43.4 % from students’ are aware that this process requires a long time and tailor-made strategies.
56.6 % percent from students don’t want to learn new information about bioplastic, maybe their fields  are different and they consider because their specialty is different and they consider that what they know is enough or the information they have through mass media promotion is enough. In a percent of 13.2% some students’ are still aware of the importance of plastic waste and consider it’s necessary to continue research in this field of removing plastic and replacing it in time. 
For item N1:‘’I would like to learn new information about the use of bioplastic products.’’ only 20.3 % percent from students from both specialities are open to learn and to obtain new information about bio plastic , maybe because some of them are already working, or they don’t want to spend extra time with new information, mention are qualify person who are already payed to do specific job. 
For item N2: ‘’Bioplastics should replace conventional polymers in the future ‘’ a 30.5% percent from students’are agree and totally agree that in future plastic must to be replace with the new material biodegradable, to prevent the pollution. 
For item N5:‘’I think that bioplastics should be used in takeaway products.’’ The biggest value of 31.5 % percent of students agrees with the bioplastic takeaway maybe because they know the importance to protect the environment and they have an environmental education already. 
But for item N4:‘’I think that bioplastics should be used in mass social events’’ the percent it’s increasing to 31.1 % so students know about the waste plastic especially from social events where they are involved. 
In conclusion students have information, knowledge’s about plastic and waste plastic, they already apply and are oriented in their daily life  to  use the new material bioplastic, but they don’t want to be involved in courses or lectures on that specific topic in their extra curricula time.
5.5. Students’ Awareness
 From awareness point of view the results for items A1-A8 from Table 5, present that students are inform and they are concerned about the issue of plastic waste and plastic pollution, but they are acting individual. 
For item A4: ‘’ I prefer the products obtained from the bioplastics industry because they are renewable’’ the biggest value of  33.9 % percent form students from both specialization management and engineering prefer bioplastic products because they are renewable, that;s mean they are inform about the waste plastic problem. 
For item A5: ‘’ I prefer to avoid products that increase global warming.’’ 32% percent from students’ agree that plastic waste problem can increase the global warning.
For item A6 ’I prefer bioplastic products because they do not harm the nature when they decompose’’ the biggest value of  33% percent from respondents agree with bioplastic because the material don’t harm the nature when they decompose.

                Table 5.  Awareness Items selection
A1     I prefer products that are less harmful to nature
A2     I use low-damage objects such as paper and glass.
A3     I recognize the bioplastic logos on the products.
A4     I prefer the products obtained from the bioplastics industry because they are renewable.
A5     I prefer to avoid products that increase global warming.
A6     I prefer bioplastic products because they do not harm the nature when they decompose.
A7     I prefer bioplastic products as they do not harm human health when degraded.
A8     I prefer bioplastic products because they degrade earlier in nature.

For item A7: ‘’I prefer bioplastic products as they do not harm human health when degraded’’ a 30.1% percent from students’ are agree and totally agree to choose the bioplastic products which are not harm health after the degraded. 
For item A8: ‘’ I prefer bioplastic products because they degrade earlier in nature’’ a value of  32,1% percent from respondents have information about the bioplastic degradation, in time and they are awareness about the environment protection, a percent of 26.1% they don’t have any opinion, they are passive.
For all items a compact value between 20 - 30% from respondents are staying and watching and consider that there are specialist payed to take care of  plastic  problem.   
5.6. Students Behavior
Students’ behavior from gender point of view,  behaviors were shown in purchasing products with plastic packaging, preparing shopping bag, re-using plastic bags, taking own meal box, and having food on the sites to reduce single used plastic package. Students’ as target it is important as these individuals will be the ones making decisions priestly and politically, with regards to recycling programs in their communities. 
By understanding their behavior in colleague and universities it is possible for those who design recycling program to develop and modify program in function if their needs and information program for their education. In Table 7 we present the students’ behavior items survey B1-B9.
For item B2:’’ When I see plastic being burned, I warn it, thinking it will cause air pollution.’’ a percent of 31.3 % of students have a positive behavior when they are involve in different plastic waste situation which can harm the nature and polluted.  

                           Table 7. Items for students ‘behavior regarding waste plastic
B1    I buy bioplastic products, even if they are expensive.
B2    When I see plastic being burned, I warn it, thinking it will cause air pollution.
B3    I use bioplastic bags for my grocery shopping.
B4    I make an effort to use a small number of bags in my daily life.
B5    I do not throw plastic products into nature.
B6    I throw the recycling products into the relevant boxes.
B7    I use mesh/cloth/paper bags instead of using disposable bags while shopping.
B8    I inform the people around me about the use of bioplastic products.
B9    When I see plastic in the green area, I take it from there.

For item B7:’’I use mesh/cloth/paper bags instead of using disposable bags while shopping.’’ a 30. 4 % percent are trying to adapt their behavior with the new standards, the low value obtained it is maybe because they are or prepare yet with new situation or maybe the cost are increasing a little bit.  
For item B8: ‘’I inform the people around me about the use of bioplastic products.’’ 33 %  percent of students’ from both specialties agree with the importance of dissemination and information regarding the use of bioplastic products.
For item B9: ‘’When I see plastic in the green area, I take it from there’’ the highest value of 33% percent of the students have a protective behavior towards nature and green nature by selecting waste in specific places.

Point 4- For clearer understanding of the steps and methods of your study it would be better to provide Research methodology section. Please revise accordingly.
Response 4. 
4. Research Methodology 
4.1. Materials and Methods
The purpose of the study was to understand the students' behavior and awareness regarding the sustainable environment and students’ behavior and knowledge with reference to the phenomenon of plastic recycling. A total of 537 students were involved in an experimental study to identify information and the level of knowledge with reference to the current problem of plastic and the need to replace it with bio plastic. 
The questionnaire was applied online between June and July 2022 at the Technical University of Cluj Napoca, Romania, from department of economics and engineering. The survey was structure in three parts: 
Part 1. Students’ characteristics (gender, faculty, specialization, participation and attending activities about environment protection and their environmental attitude (I1-I7 questions );
Part 2: Students’ awareness (A1-A8 questions) to examine whether the young generation plays a responsible role towards plastic products and their perception towards this topic plastic and new biodegradable plastic;
Part 3. Students’ needs (N1-N5 questions) perception of the concept of plastic sustainability and their participation in different activities regarding plastic recycling
Part 4. Students' behavior (B1-B9 questions) about healthy plastic education,  how many of them select plastic, if they use ecological products. 
In this study, we’re used Google Drive form to create the survey, and apply online, data analyze and statistical processing were performed using the SPSS software package and for students’ model Rigle et al.[59], Starstedt et al.[60] and Hair et al.[61] SmartPLS 3 program.

4.2.    Sample and Measurement Tool
To measure the student’s behavior, needs and awareness on waste plastic, a Likert-scale type questionnaire, ranging from 1 ‘Totally Appropriate’ to 5 ‘Not at all appropriate’, was applied. Research model is given in Figure 1 we’re taken into consideration: N- Needs; A-awareness about environment and plastic and B-Behavior. 

Figure 1. Research model regarding plastic waste. Source: By authors

In this study, the following factors were taken in consideration: 
 B- students behavior (B1, B2, B3, B4, B5, B6, B7, B8 and B9 questions like in Table 4). 
A- students awareness regarding waste plastic ( A1, A2, A3, A4, A5, A6, A7 and A8 questions like in Table 5) ;
N- students needs ( N1, N2, N3, N4 and N5 questions like in Table 6);
To determine the dimensions of students’ behavior and identify students’ needs and awareness about waste plastic and bioplastic an Explanatory Factor Analysis (EFA) were applied to the data set. By independent samples t test, the hypotheses were tested using the SPSS statistical analyses software.
Hypothesis for the model given in  Figure 1 are as below:
H1: Students Awareness towards plastic waste effect their Needs;
H2: Students Awareness towards plastic waste effect their Behavior;
H3: Students Needs toward plastic waste effect their Behavior.

Point 5. The conclusion section is too length and does not reveal the main results of the study and their practical implication. 
Response 5. I made  the correction. Rewritten Conclusion
The results obtained after filling out the questionnaire led to the creation of the new model and identifying common elements of students’ needs, students’ behavior and students’awareness regarding waste plastic. 
For our study Romanian students’ awareness obtain the highers value so they are involve in waste plastic management. 
We also verified, the influence of the profile to which the students are enrolled in our case the management profile students knowledge s are higher from theoretical point of view but engineering students are more practical, and they are involved in activities and active participation organized by university or they are attending conferences on plastic topic.  
Authors identify the similarities and differences when we take in consideration students’ gender, male are more practical in cleaning  the nature from plastic, in comparison with female students which pay attention to small details of using, selecting the waste plastic. 
Students’ are acting individual as person and they don’t like to be involve in extra activities. They are inform from mass media, from university but sometimes they are lazy to follow the rules or they consider that are qualify person who are payed to do their job. The results are a signal for universities which must to involve students in activities with specific topic about environment protection and waste plastic, in volunteer activities and also in research activities to understand the importance of subject. In Romania, efforts to combat the plastic pollution crisis are rather minor, fragmented, and generally ignore the extremely harmful consequences that both  we and the environment have to bear every day. 
Also on the study of bio plastic, and the future vision without plastics present also that the respondents and also the organization must to be careful when it comes to alternative plastic (cane or corn starch product, biodegradable and / or compostable) and analyze all the sustainability issues associated with this type of plastic. 
For the compact segment identified in the research study of 20-30 % percent of the respondents who have no opinion and do not want to be involved in any kind of activities just sit, we can specify the following factors: 
•    the lack of concern for this problem, first of all by the shops and then by authorities who do not properly inform about the problem of plastic;
•    lack of information is another reason specified by the respondents for the lack of information time .
The Cross Model for students in relation to bio plastic obtained a low value, which means  a weak connection between the needs of students that influence their behavior. Connection needs present a weak point in the education of students regarding environmental protection with plastic issues as a direct target. So a necessity for students’ needs, in their behavior towards sustainable environment education is influence by knowledge.  
Precisely , the factors that led to this disastrous situation are the following: 
•    lack of a separate collection system at source (at people's homes) on at least 4-5 types of waste: paper / cardboard, plastic / metal, glass , bio waste (food and vegetable waste) and mixed; 
•    lack of proper education among the population and decision-makers at central and local level;
•    lack of involvement of mayors and failure to assume their responsibilities;
•    the lack of sanctions and total lack of waste collection in rural areas, where there are no sanitation contracts, which leads most often either when disposing of them in the wild or when burning uncontrolled.
The strongest link is the one between the students' awareness influences their information needs regarding the plastic problem. This is where the role of universities comes into play by: 
•    creating platforms for courses specific to the field, creating virtual libraries as a source of useful information;
•    the transfer of information between different universities or student groups for common research topics.
The students' awareness influences their behavior by getting involved in voluntary actions, through the individual actions of selecting plastic at home or at university, protecting nature by choosing biodegradable products when shopping.
Point 6 .Discussion section can be added.
Response 6. I added
Point 7. Professional proofreading is required. I found many typos and grammatical errors in sentence structures.
Response 7: We made  the correction. The article will be verify by the professional personal from MDPI 
Point 8- For some statements, you need to cite their references. For instance, in the introduction section “In order to join Europe, a stand-alone place was occupied by Chapter 22 "Environmental protection", and within it the position "Waste" was noted”. You should refer to the relevant reference. 
Response 8: I made  the correction.
Sustainable development has a key component to environmental issues which is one of the Union's horizontal policies European.  In order to join Europe, a stand-alone place was occupied by Chapter 22 "Environmental protection", and within it the position "Waste" was noted [1]. 
1.    Plastic waste and recycling in the EU: facts and figures | NewsWaste recycling - European Environment Agency, Available online: https://www.europarl.europa.eu/news/en/headlines/society/20181212STO21610/plastic-waste-and-recycling-in   -the-eu-facts-and-figures (accessed on 1 February 2023). 
Point 9. The same comment for this sentence: “Germany and Italy are managing very well the management of recycling plastic in comparison with Romania and Finland which occupied the last place”. 
Response 9: I delete the paragraph. Too many  details 
Point 10. The same comment for this sentence: “Romania recycles only 13% of waste, and the rate of disposal of waste by final disposal at landfills is 69%, among the highest in Europe”. 
Response 10: I made the correction 
 Romania recycles only 13% of waste, and the rate of disposal of waste by final disposal at landfills is 69%, among the highest in Europe [5].
3.    Studiu: Alternative pentru o Românie fără plastic – percepții și comportamente ale românilor, Available online: https://www.
      invisiblenature.ro/sustainability/studiu-alternative-pentru-o-romanie-fara-plastic/(accessed on 28 February 2023).

Point 11. There are so many other statements that need reference support. This is not acceptable in academia and you must carefully check such issues when writing a paper. Please check this issue within the whole manuscript.
Response 11: I made the correction in text and put to References.
Statistics showed that almost a third of plastic waste is recycled in Europe plastic production has grown exponentially worldwide in just a few decades, from 1.5 million tons in 1950 to 359 million tonnes in 2018 [2]. 
After a sharp decline in the first half of 2020 due to Covid-19, production plastic recovered in the second half of the year and with that came plastic waste [2] but EU is already taking steps to reduce the amount of plastic waste. Androniceanu et al.[3]  estimate that in 2019, globally, the production and incineration of plastics released more than 850 million tons of greenhouse gases into the atmosphere and by 2050, these emissions could reach 2.8 billion tons. Some of these can be avoided by better recycling. 
Unfortunately, Romania is among the 14 EU member states that risk not being able to meet the 50% waste recycling target in 2020. The European Commission's analysis shows that the main causes of the problem are related to the fact that [4]: Although recycling cannot replace the need to significantly reduce the amount of disposable packaging and is by no means a justification for increasing plastic production, it has an important role to play in the transition to a plastic-free economy [6]. 
Bastos de Sousa [7] mentioned the essential role of the Agenda 2030 for Sustainable Development [8], which sets the Sustainable Development Goals and provides a critical overview of the role of plastic, mentioning its advantages and disadvantages.
Added to References 
2. Plastic waste and recycling in the EU: facts and figures, Available online: https://www.europarl.europa.eu/news/ro/ headline
     s/society/20181212STO21610/deseurile-din-plastic-si-reciclarea-in-ue-in-cifre ( accessed on 13 February 2023)
4.    Androniceanu A; Kinnunen, j.; Georgescu, I., Circular economy as a strategic option to promote sustainable economic growth and effective human development,Journal of Internatinal Studies, 2021, 14(1) 60-73
5.    Legea anti-plastic - mai bine pentru mediu, mai provocator pentru firme, Available online: https://www2.deloitte.com/ro/ro/ 
      pages/tax /articles/legea-anti-plastic-mai-bine-pentru-mediu-mai-provocator-pentru-firme.html (accessed on 16 February 2023). 
6.    Studiu: Alternative pentru o Românie fără plastic – percepții și comportamente ale românilor, Available online: https://www.
      invisiblenature.ro/sustainability/studiu-alternative-pentru-o-romanie-fara-plastic/(accessed on 28 February 2023).
7.    Pentru un viitor nesufocat de plastic, Available online: https://www.greenpeace.org/romania/articol/4507/pentru-un-viitor
    -nesufocat-de-plastic/( accessed on 28 Februarie 2023) 
Point 12- In the introduction section the authors have written “Our country has committed itself to…”, but they do not name any country. Please pay attention that you have submitted your article to an international journal, and you should be more careful with some amendments in this regard.
Response 12: I made  the correction.
Romania has committed itself to a substantial reduction in a period of 15 years quantities of waste, to make the necessary systems for collection, recycling  and revaluation, but also those of future protection of the environment and a population against pollution.
Point 13- In the literature review section, there are many short paragraphs that should be linked with each other. Please try to incorporate some of the paragraphs to provide a better and more solid flow of information. Besides, you can refer to the significant effect of the pandemic on plastic waste generation and management. In this regard, you might refer to the paper titled “waste management beyond the covid-19 pandemic: bibliometric and text mining analyses”.
Response 13: I introduce in text   
A significant effect of the pandemic on plastic waste generation and management was register. Prasetiawan and Wasisto[24] investigated the relationship between students' perception and attitudes in waste management using the Internet leads to changes in attitude and behavior in society. Several studies have also highlighted the positive impact of COVID-19 on the environment, while limited published studies the long-term negative impact of COVID-19 on the environment and waste management. A negative aspect was observed by Ranjbari et al.[25] and Mohamed et al.[26] because during the COVID-19 pandemic there was a significant increase in the demand for personal protective equipment, face masks, increasing globally and the amount of plastic waste in the health sector. The effects of the COVID-19 pandemic were also felt on the production, use and waste of plastic materials, on economic activities and the use of plastic materials. The use of plastic materials decreased in the industrial sectors, as did the consumption of plastic materials in construction, the automotive industry decreased. The pandemic has led to an increased demand for single-use plastic, increasing the pressure on this already out-of-control problem. Peng et al.[27] noted that plastic waste also harms marine life and has become a major environmental concern worldwide.
Point 14- Please avoid reference lumping. For each statement, 3 references are adequate. Otherwise, you should separate the content. For instance, in line 285, you have cited [7], [10], [11], [17], [19-20], [22-23], [36-37], [45], all together.
Response 14: I made  the correction. New version 
As a conclusion until now, the main focus of research has been on high school, college, university students around the world on the problem of waste plastic, on waste management, or recycling and reuse of plastic, and the attitude of students towards the new biodegradabile plastic [11], [24], [29], students’ behavior towards the environment [19], [46], students’ behavior towards plastic waste [20], [22], [24], students’ knowledge about plastic pollution [28], [42], [45]], students’ knowledge about the plastic waste [9], [11], [30], students’ awareness about plastic [19], [46]  and students’ needs [44], [47]. Environmental protection was raised to the rank of objective of major public interest and responsibility, based on the fundamental principles of environmental law: the principle of conservation, the principle of prevention, the principle of precaution in decision-making and the polluter pays. Taking in consideration that literature has so far focused on specific aspects of plastic waste and sustainability in the higher education sector, the aim of this study is to explore the impact toward of Romanian students’.

Point 15. Besides, I did not understand why the authors presented section 
6.1 “A SWOT Analyze for Romanian students” in the conclusion section. Why here??? Should you not present this above the conclusion? 
I recommend revisiting this section. In addition, the sections are too short, and I think there is no need to name the sub-sections. For instance, section 6.2 is only 2 sentences.
 Response 15: I put the SWOT Analyze in Discussion section.

Round 2

Reviewer 1 Report

I appreciate the efforts authors put into revising the manuscript. The quality of the paper has now improved and I can recommend it for publication. However, there is one more minor point that should be considered in the text. I could not find the full form of SWOT before using its abbreviation. Please introduce the full form before using SWOT. Please also check this issue for all abbreviations throughout the manuscript.

Author Response

Response to Reviewer 1 Comments

Dear Editor

Dear Reviewer

I very much appreciated the encouraging, critical and constructive comments on this manuscript by the reviewer. The comments have been very thorough and useful in improving the manuscript. I strongly believe that the comments and suggestions have increased the scientific value of revised manuscript by many folds. I have taken them fully into account in revision. I am submitting the corrected manuscript with the suggestion incorporated the manuscript. The manuscript has been revised as per the comments given by the reviewer, and our responses to all the comments are as follows:

I appreciate the efforts authors put into revising the manuscript.

The quality of the paper has now improved and I can recommend it for publication.

 However, there is one more minor point that should be considered in the text.

Point 1. I could not find the full form of SWOT before using its abbreviation. Please introduce the full form before using SWOT.

Response 1: We made  the correction.

In conclusion, for our study in Table 7 we create a Strengths, Weaknesses, Opportunities, and Threats matrix know as S.W.O.T. analysis  related to understand  our strong and weak points and what we have to do for students.

Point 2.  Please also check this issue for all abbreviations throughout the manuscript.

Response 2. Done

Reviewer 2 Report

Article substantially revised and improved according to the comments. 

I recommend to publish the paper in Substantially.

Author Response

Response to Reviewer 2 Comments

Dear Editor

Dear Reviewer

I very much appreciated the encouraging, critical and constructive comments on this manuscript by the reviewer. The comments have been very thorough and useful in improving the manuscript. I strongly believe that the comments and suggestions have increased the scientific value of revised manuscript by many folds. I have taken them fully into account in revision. I am submitting the corrected manuscript with the suggestion incorporated the manuscript. The manuscript has been revised as per the comments given by the reviewer, and our responses to all the comments are as follows:

Point 1.

Article substantially revised and improved according to the comments. 

I recommend to publish the paper in Substantially.

Response 1: We really appreciate

Reviewer 3 Report

Please refer to word document. Make sure to submit to official proofreading only then the journal will consider the acceptance

Author Response

Response to Reviewer 3 Comments

Dear Editor

Dear Reviewer

I very much appreciated the encouraging, critical and constructive comments on this manuscript by the reviewer. The comments have been very thorough and useful in improving the manuscript. I strongly believe that the comments and suggestions have increased the scientific value of revised manuscript by many folds. I have taken them fully into account in revision. I am submitting the corrected manuscript with the suggestion incorporated the manuscript. The manuscript has been revised as per the comments given by the reviewer, and our responses to all the comments are as follows:

Point 1. Please refer to word document. Make sure to submit to official proofreading only then the journal will consider the acceptance

Response 1: We made  the correction.

Point 2. Title Effects of Romanian student’s awareness and needs regarding waste plastic management, suggest to revise to plastic waste management

Response 2. We have changed 

Effects of Romanian student’s awareness and needs regarding plastic waste management

Point 3.  Abstract Their awareness regarding plastic waste depends on their needs and behavior. Also their needs influence their behavior, this is the example of hanging sentences, suggest to revamp the whole abstract to make it more coherent.   

Response 3.

Abstract:. The purpose of this study is to examine the effects of nedds and aweraness of university students on their environmental behaviour. With this purpose the data was collected from 537 students from the University of Cluj Napoca, Romania, from the engineering and management specializations respectively via an online questionairee. The  questionnaire was was structured in four parts including 29questions in total. The first part is meant to identify the students’ characteristics (gender, field of  study, participation and attendance in field-specific activities , and if he/she is an environmentalist). The second part is meantto determine the students’ awareness reagarding plastic and plastic pollution. Another part is meant to determine the needs of students and the  manner in which they learn and gather information. The last part allows the determination of the students’ behavior in their daily life (use of bio plastic bags, environmental protection). The results show that students have enough information about biodegradable plastic but they act depending on the situation, respecting or not the rules for selecting plastic waste. The female student’  pay a lot of attention  to selecting and choosing bioplastic products.The male students are directly involved in cleaning nature. Management students pay attention to small details as compared to engineering students who choose bioplastic even though the costs are higher. Related with their thoughts the factors effecting the opinion of either they are environmentalist or not are also examined. Being aware of the Plastic waste show significant effect from the sides of awereness and behaviour. Finaly the stuructural model  show that strongest connection is between students’ awareness about the plastic problem and the need to adapt to new regulations. Their awareness regarding plastic waste depends on their needs and behavior. Also their needs influence their behavior.  Using the model universities can promote the importance of bioplastic through study programs or by involving students in volunteering activities, through their active involvement in environmental protection, and selective waste recycling.

Romanian student’s behavior, needs and awareness regarding Plastic Waste